# Constraints on post-depositional isotope modifications in East Antarctic firn from analysing temporal changes of isotope profiles

Thomas Münch[1,2], Sepp Kipfstuhl[3], Johannes Freitag[3], Hanno Meyer[1], and Thomas Laepple[1]

[1]Alfred Wegener Institute Helmholtz Centre for Polar and Marine Research, Telegrafenberg A43, 14473 Potsdam, Germany
[2]Institute of Physics and Astronomy, University of Potsdam, Karl-Liebknecht-Str. 24/25, 14476 Potsdam, Germany
[3]Alfred Wegener Institute Helmholtz Centre for Polar and Marine Research, Am Alten Hafen 26, 27568 Bremerhaven, Germany

*Correspondence to:* Thomas Münch (thomas.muench@awi.de)

**Abstract.** The isotopic composition of water in ice sheets is extensively used to infer past climate changes. In low-accumulation regions their interpretation is however challenged by poorly constrained effects that may influence the initial isotope signal during and after deposition of the snow. This is reflected in snow-pit isotope data from Kohnen Station, Antarctica, which exhibit a seasonal cycle but also strong inter-annual variations that contradict local temperature observations. These inconsistencies persist even after averaging many profiles and are thus not explained by local stratigraphic noise. Previous studies have suggested that post-depositional processes may significantly influence the isotopic composition of East Antarctic firn. Here, we investigate the importance of post-depositional processes within the open-porous firn ($\gtrsim 10\,\mathrm{cm}$ depth) at Kohnen Station by separating spatial from temporal variability. To this end, we analyse 22 isotope profiles obtained from two snow trenches and examine the temporal isotope modifications by comparing the new with published trench data extracted 2 years earlier. The initial isotope profiles undergo changes over time due to downward-advection, firn diffusion and densification in magnitudes consistent with independent estimates. Beyond that, we find further modifications of the original isotope record to be unlikely, or small in magnitude ($\ll 1\permil$ RMSD). These results show that the discrepancy between local temperatures and isotopes most likely originates from spatially coherent processes prior to or during deposition, such as precipitation intermittency or systematic isotope modifications acting on drifting or loose surface snow.

## 1 Introduction

The isotopic composition of water measured in firn and ice cores is an important climate proxy. The abundance ratios of the stable water isotopologues in falling snow are shaped by different fractionation processes in between the moisture source and the precipitation site, including evaporation (Craig and Gordon, 1965), air-mass advection and Rayleigh distillation (Dansgaard, 1964), and snow formation (Jouzel and Merlivat, 1984). Hence, isotope ratios can be linked to the climatic conditions at the local or moisture source site. For instance, physical modelling of the large-scale hydrological cycle and the fractionation processes has validated the link between the isotopic composition of precipitation and local temperature (Jouzel et al., 1997, 2003, and references therein) previously inferred for polar ice sheets, where observational evidence has suggested a robust relationship at large spatial scales (i.e. continental) between the isotopic composition of snow and annual-mean temperature

at the sampling sites (Dansgaard, 1964; Lorius et al., 1969; Masson-Delmotte et al., 2008). Isotope data archived in polar ice cores have therefore become an invaluable means to infer past site temperature variations (e.g. Petit et al., 1999; NEEM community members, 2013) or changes in the moisture sources (e.g. Vimeux et al., 2001; Uemura et al., 2012), and show, at least qualitatively, a globally consistent picture of glacial–interglacial to millennial-scale climate changes (EPICA community

members, 2004, 2006; NGRIP members, 2004). However, it is questioned whether the assumption that pre-depositional fractionation processes alone are the main influence on the isotopic composition of firn and ice, while seemingly fulfilled for large spatial and temporal scales, holds in general. Particularly in low-accumulation areas for which the snow surface is exposed to the atmosphere for a substantial time, a variety of processes are thought to considerably modify the original atmospheric isotope signal during or after deposition of the snow, thus from seasonal to inter-annual timescales (e.g. Ekaykin et al., 2014,

2016; Hoshina et al., 2014; Touzeau et al., 2016; Casado et al., 2016).

For the East Antarctic Plateau, modifications of the original isotope signal that is imprinted in precipitation are generally exptected. In buried snow and firn, the isotopic composition is affected by diffusion of interstitial water vapour (Johnsen, 1977; Whillans and Grootes, 1985; Cuffey and Steig, 1998; Johnsen et al., 2000; Gkinis et al., 2014) and by densification (Hörhold et al., 2011, 2012; Freitag et al., 2013b); however, these processes do not lead to any net change in the isotopic composition.

In contrast, the seasonal intermittency of precipitation and accumulation can bias the original signal, induce variability, or lead to a combination of both (Sime et al., 2009, 2011; Persson et al., 2011; Laepple et al., 2011). In combination with the low accumulation rates on the East Antarctic Plateau, precipitation intermittency also increases the time the surface is exposed to the atmosphere (Town et al., 2008; Hoshina et al., 2014). These conditions might favour fractionation, diffusive and advective processes that can considerably alter the snow's original isotopic composition, acting either post condensation (on falling or

drifting snow), or post-depositionally on snow at the surface or within the open-porous firn column which is no longer subject to erosion but still in contact with the atmosphere. For instance, exchange of water vapour between the first metre of firn and the overlying atmosphere through diffusion and wind-driven ventilation (Waddington et al., 2002; Neumann and Waddington, 2004; Town et al., 2008) can introduce vapour with a different isotopic signature to the firn and significantly change the isotopic composition. Isotopic exchanges between the top layer of snow and the lower atmosphere have been observed on daily scales

at the NEEM site in Greenland (Steen-Larsen et al., 2014) and on diurnal scales at Kohnen Station in East Antarctica (Ritter et al., 2016). Isotopic fractionation associated with sublimation, condensation and recrystallisation processes within the near-surface firn might change the initial isotope signal, as indicated by observations (Moser and Stichler, 1974; Stichler et al., 2001) and lab experiments (Hachikubo et al., 2000; Sokratov and Golubev, 2009). Since these post-depositional processes depend, besides temperature, also on other climatic variables such as wind speed and relative humidity, any seasonal or inter-

annual variations in these variables would induce additional variability in the isotope record. However, for East Antarctica, a quantitative assessment of the individual processes based on firn-core data is still outstanding, and their importance for shaping the isotope signal in the near-surface firn remains poorly constrained.

An additional, important source of variability in low-accumulation firn-core records is the spatial variability from stratigraphic noise (Fisher et al., 1985), caused by uneven deposition and the constant wind-driven erosion, redistribution and

vertical mixing of the snow surface. A previous study from Kohnen Station in Dronning Maud Land, East Antarctica, has

shown that the spatial variability can be overcome by averaging across a suitable number of single profiles extracted from snow trenches (Münch et al., 2016). This yielded a spatially representative isotope signal on a horizontal scale of approximately $500\,\mathrm{m}$. However, contrasting the isotope data with instrumental observations from a nearby automatic weather station (AWS, Reijmer and van den Broeke, 2003) suggests that this regional signal does not necessarily represent a regional temperature

signal (Fig. 1). Whereas the isotope record shows strong year-to-year variability, the observed temperature variations are characterised by a regular seasonal cycle and small inter-annual changes. This discrepancy stresses the importance of contributions other than regional temperature alone to the formation of the isotope signal, such as precipitation intermittency and changes during or after deposition. Since quantitative knowledge on seasonal and inter-annual variations of accumulation amounts is still sparse on the East Antarctic Plateau (Reijmer and van den Broeke, 2003; Helsen et al., 2005), in this study we investigate

whether post-depositional isotope modifications in the open-porous firn contribute to the observed discrepancy between the isotope data and local temperatures at Kohnen Station.

One way to address the question of post-depositional modification is to compare two firn-core isotope profiles obtained at different times and to measure the nature in which the first profile has been modified. However, due to stratigraphic noise, the comparison of two single records sampled at different times will always confound temporal isotope changes and spatial

variability. Therefore, in this study we present and use data from a new extensive snow trench campaign yielding 22 profiles of isotopic composition from two trenches, and compare these with the data of the previous trench campaign conducted 2 years earlier. By generating representative records from the spatial averaging of single profiles, together with the theoretical understanding of stratigraphic noise, our study allows us for the first time to quantitatively follow the isotopic changes over a time span of 2 years. We use independent knowledge on firn diffusion and densification to subtract these effects from the observed

temporal modifications. Therefore, beyond simply stating the problem of local isotope–temperature discrepancy, we go further and can quantitatively estimate the influence of post-depositional change for our study site. This is an important step towards better constraining the isotope signal formation in East Antarctic firn.

## 2   Data and methods

### 2.1   Sampling and measurements

A pair of firn trenches, each with a horizontal length of $50\,\mathrm{m}$ and a depth of $3.4\,\mathrm{m}$, was excavated using a snow blower in the austral summer field season 2014/2015 near Kohnen Station (Alfred-Wegener-Institut Helmholtz-Zentrum für Polar- und Meeresforschung, 2016), the location of the EPICA Dronning Maud Land deep ice core drilling site (Fig. 2 and Table 1). This campaign extends the published oxygen isotope data set obtained from two shallower ($\sim 1\,\mathrm{m}$) trenches in 2012/2013 (Münch et al., 2016). From the new trenches, we present the top $1.75\,\mathrm{m}$ of the data which are expected to cover the period imprinted in

the trenches of the first campaign. To avoid contamination, the new trench positions were shifted relative to the previous ones by $160\,\mathrm{m}$ and $300\,\mathrm{m}$, respectively, and are separated by $550\,\mathrm{m}$ (Fig. 2). In the remaining part of the manuscript, "T13" will refer to the pair of previous trenches from 2012/2013, "T15" to the pair of new trenches from 2014/2015.

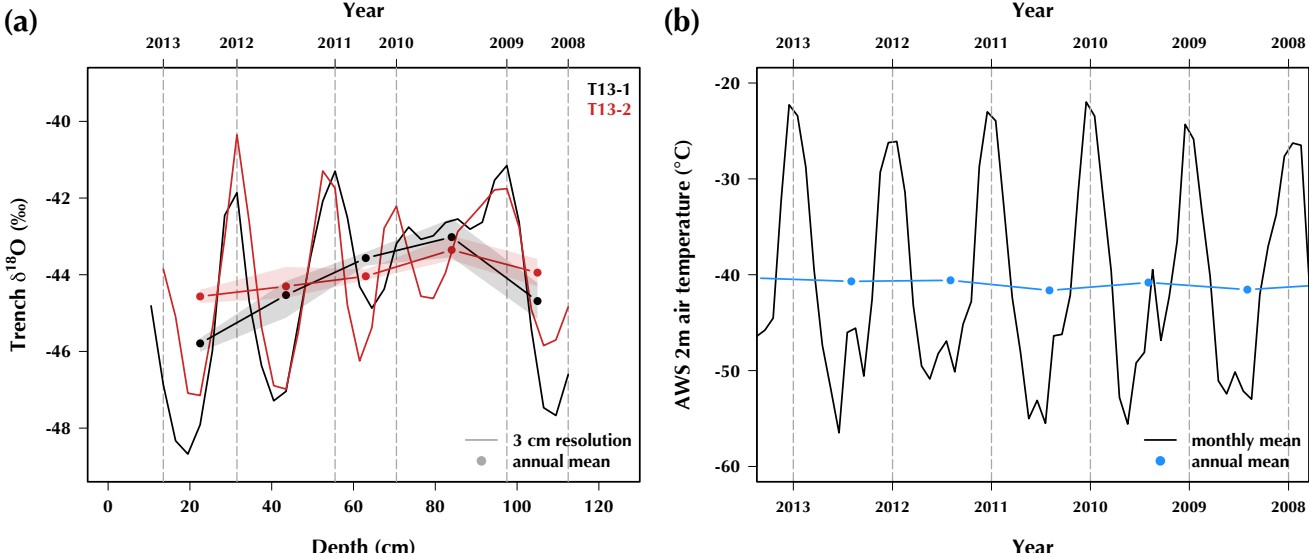

**Figure 1.** Comparison of oxygen isotope data and 2 m air temperature at Kohnen Station, Antarctica. (**a**) Mean $\delta^{18}$O profiles of trenches T13–1 (black) and T13–2 (red) (modified after Münch et al. (2016)) on original 3 cm (lines) as well as annual resolution (points with uncertainty shading from shifting the range of the annual bins). (**b**) 2 m air temperature (black lines: monthly means, blue points: annual means) recorded by the automatic weather station AWS9 located at Kohnen Station $< 1$ km from the trench excavation sites. Note the different timescales (**a**: based on counting and binning the extrema of the isotope data, **b**: true calendar dates). The mean of the 2 m air temperature shown here lies about 3.5 °C above the published local 10 m firn temperature (Table 1).

**Table 1.** Information on the EPICA Dronning Maud Land (EDML) drilling site at Kohnen Station, Antarctica. Listed are approximate position (latitude, longitude), elevation, 10 m firn temperature $\overline{T}_{\text{firn}}$, mean annual accumulation rate of snow $\overline{b}$, and mean daily wind speed $\overline{v}_{\text{wind}}$ ($\pm 1$ standard deviation).

| Drilling site | Latitude | Longitude | Elevation | $\overline{T}_{\text{firn}}$ | $\overline{b}$ | $\overline{v}_{\text{wind}}$ |
|---|---|---|---|---|---|---|
| | °N | °E | m a.s.l. | °C | mm w.eq. yr$^{-1}$ | m s$^{-1}$ |
| EDML | $-75.0$[a] | $0.1$[a] | 2892[a] | $-44.5$[a] | 64[a] / 82.5[b] | $4.4 \pm 2.3$[c] |

[a] EPICA community members (2006). [b] Mean of snow stake measurements 2013–2015. [c] AWS9 data 1998–2013 (Reijmer and van den Broeke, 2003).

Fieldwork for the new T15 trench campaign was conducted as follows: Horizontal profiles of the surface height variations were obtained along each trench using a levelling instrument. The uncertainty of these profiles is estimated from the reading accuracy of the levelling rod of 0.5 cm. The windward walls of the trenches were cleaned after excavation by slicing off a thin firn layer. Firn profiles were then sampled directly off the wall with a vertical resolution of 3 cm and a horizontal spacing of 5 m, yielding 11 profiles in each trench. The vertical resolution is small enough to evaluate the seasonal cycle of the istope data

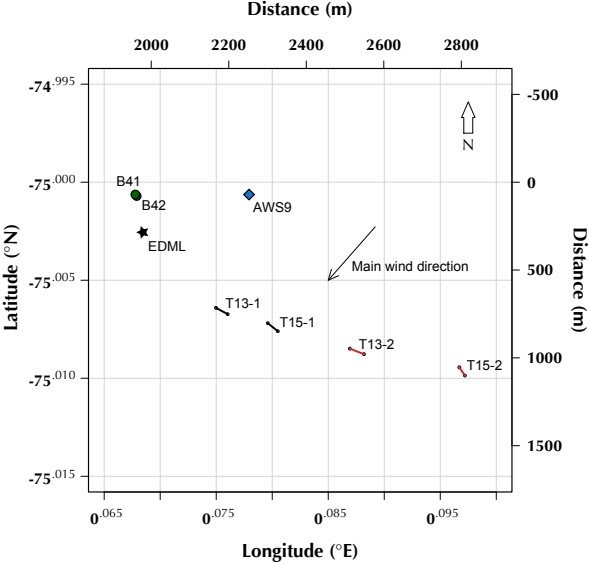

**Figure 2.** Map of the study area at Kohnen Station. Snow trenches are shown as black and red lines, firn-core sites as green filled circles. The drilling site of the EPICA Dronning Maud Land (EDML) ice core is shown as a black star, the position of the automatic weather station (AWS9) as a blue filled diamond. The main wind direction (57° from geographic North, Birnbaum et al., 2010) is indicated with a black arrow. The trenches were aligned perpendicularly to the local snow-dune direction.

of $\sim 20\,\text{cm}$ (Münch et al., 2016); the inter-profile distance of $5\,\text{m}$ corresponds approximately to three times the decorrelation length of the stratigraphic noise observed in the T13 record (Münch et al., 2016). At both trenches, excavation and subsequent sampling of the profiles was conducted in two consecutive stages (2 times $\sim 1\,\text{m}$ depth); each stage was completed within 24 h. All firn samples ($N = 1214$) were stored in plastic bags, tightly packed, transported to Germany in frozen state and analysed

5   for oxygen ($^{18}\text{O}/^{16}\text{O}$) and hydrogen ($^{2}\text{H}/^{1}\text{H}$) isotope ratios at the isotope laboratory of the Alfred Wegener Institute (AWI) in Potsdam, using a cavity ring-down spectrometer (L2130i, Picarro Inc.). The results are reported in the usual delta notation (oxygen isotopes: $\delta^{18}\text{O}$, hydrogen isotopes: $\delta\text{D}$) in per mil (‰) relative to the international V-SMOW/SLAP scale. Calibration and correction of the raw measurements was performed as described in Münch et al. (2016). The mean combined measurement uncertainty is $0.08\,‰$ for $\delta^{18}\text{O}$ (root-mean square deviation (RMSD)) and $0.8\,‰$ for $\delta\text{D}$, assessed by evaluating a standard not

10   used in the calibration and correction procedure.

## 2.2   Trench depth scale

Following Münch et al. (2016), we record and display the trench isotope data with respect to an absolute height reference given by the respective maximum of the surface height profile of each trench. Note that the surface layer of the trench records is incomplete on this depth scale (up to $\sim 10\,\text{cm}$ for T13 and $\sim 18\,\text{cm}$ for T15) due to the surface undulations. Averaging of

**Table 2.** Sampling and statistical properties of the trench $\delta^{18}$O records from the field seasons 2012/2013 (T13, Münch et al., 2016) and 2014/2015 (T15, this study). Listed are: Number and distance of sampled profiles, $\delta^{18}$O values and variance, correlation of mean trench profiles, and estimated signal-to-noise variance ratios (SNR) after Münch et al. (2016). Correlations are maximised through allowing relative vertical shifts (optimal shift given in brackets). 67% confidence intervals (CI) for the variance estimates account for autocorrelation of the data. Average signal-to-noise ratios are given with an uncertainty of 1 standard error (SE).

| Trench record: | T13–1 | T13–2 | T15–1 | T15–2 |
|---|---|---|---|---|
| number of profiles: | 38 | 4 | 11 | 11 |
| profile distances (m): | $\sim 0.1$–2.5 | 10, 20 | 5 | 5 |
| $\delta^{18}$O (‰): | | | | |
| range: min/max | $-54/-34$ | $-50/-38$ | $-56/-32$ | $-55/-33$ |
| mean (SD) | $-44.4\,(3.1)$ | $-44.0\,(2.7)$ | $-44.7\,(3.8)$ | $-44.5\,(3.8)$ |
| $\delta^{18}$O variance ($(‰)^2$): | | | | |
| mean horizontal (67% CI) | $5.9\,(5.2$–$7.0)$ | $5.3\,(4.2$–$7.0)$ | $7.0\,(6.1$–$8.3)$ | $6.6\,(5.7$–$7.7)$ |
| mean vertical (67% CI) | $9.5\,(8.3$–$11.1)$ | $7.3\,(5.9$–$9.6)$ | $13.8\,(12.0$–$16.3)$ | $14.2\,(12.3$–$16.8)$ |
| mean profile correlation (optimal shift) | 0.81 (+3 cm) | | 0.91 ($-0.5$ cm) | |
| SNR ($\pm 1$ SE) | $0.9 \pm 0.4$ | $0.5 \pm 0.5$ | $1.0 \pm 0.3$ | $1.5 \pm 0.5$ |

trench profiles is performed relative to the absolute height reference. Therefore, the number of data points contributing to a mean profile is lower and varies in the surface layer. This part is marked by dashed lines for all mean profiles and is excluded from all quantitative calculations. Our conclusions are therefore limited to firn depths below $\sim 10$ cm but are however, as will be shown, not essentially changed when including the surface layer.

## 2.3 Spatial variability of average trench profiles

Spatial variability arising from stratigraphic noise is a major contribution to the overall variability of individual trench isotope profiles (Münch et al., 2016). Its magnitude $\omega$ can be estimated from the horizontal variability of the trench isotope record. Averaging across individual trench profiles reduces the total noise variability. Specifically, stratigraphic noise can be modelled by a first-order autoregressive process with a horizontal decorrelation length for the study region of $\lambda \simeq 1.5$ m (Münch et al., 2016). Then, the residual noise variability of a mean profile built by averaging across $N$ individual records is

$$\varepsilon_{\text{res}} = \frac{\omega}{N^2} \left( N + f(N, d, \lambda) \right) \equiv \frac{\omega}{N_{\text{eff}}}, \tag{1}$$

where $f(N, d, \lambda)$ is a function of $N$, $\lambda$ and of the inter-profile distances $d$. Eq. (1) equivalently can be expressed through the effective number of records, $N_{\text{eff}}$. For independent noise (zero autocorrelation, $\lambda \to 0$), $f(N, d, \lambda \to 0) \to 0$ and thus $N_{\text{eff}} \to N$.

## 2.4 Quantification of downward-advection, firn densification and firn diffusion

We expect that within 2 years the original T13 isotope profiles have been compressed through densification of the firn, advected downwards due to new snow fall and affected by firn diffusion. To quantify these effects, certain site-specific parameters have to be known.

Firn densities are provided independently of the trench data by high-resolution X-ray Computer Tomography data (Freitag et al., 2013a) of the firn cores B41 and B42 (core distance $\sim 10\,\mathrm{m}$, Laepple et al. (2016)) drilled in vicinity to the trenches ($\sim 1\,\mathrm{km}$, Fig. 2). The average firn density in the first metre is $\sim 330\,\mathrm{kg\,m^{-3}}$. The densification rate relative to the surface is $\sim 2\,\%\,\mathrm{m^{-1}}$ when regressing density against depth over the first $2\,\mathrm{m}$, $\sim 7\,\%\,\mathrm{m^{-1}}$ when regressing over the first $5\,\mathrm{m}$.

The local annual accumulation rate of snow was $28.8 \pm 0.4\,\mathrm{cm}$ ($\pm 1$ standard error) in the year 2013 and $20.8 \pm 0.3\,\mathrm{cm}$ in 2014, which was estimated from an array of snow stake measurements conducted near the trench excavation sites. In general, the recent local accumulation rate strongly depends on the measurement site, with values ranging from 20–30 cm of snow per year which is up to $50\,\%$ larger than the published longtime mean (Table 1).

In case of isothermal firn, diffusion of interstitial water vapour leads to overall smoothing of an isotope profile which can be described as the convolution with a Gaussian kernel (Johnsen et al., 2000). The amount of smoothing (the width of the Gaussian convolution kernel) is controlled by the diffusion length $\sigma$ which increases monotonically in the upper firn layer (Johnsen et al., 2000). We model $\sigma$ according to Gkinis et al. (2014) with diffusivity after Johnsen et al. (2000). Firn density is a main input to the depth dependency of the diffusion length. For the calculations we smooth the stacked B41/B42 density data by fitting a quadratic polynomial in the square root of the depth. For the concept of differential diffusion, we consider a firn layer which is located at the average depth $z_1$ and has thickness $\Delta z$ over which the increase in diffusion length ($\Delta \sigma$) is small compared to the layer thickness, $\Delta\sigma/\Delta z \ll 1$. Now the firn layer is advected downwards to the depth $z_2$. The total amount of diffusion that acted since the layer has been at the surface is the convolution of the layer's initial isotope profile at the surface, $\delta_0$, with a diffusion length $\sigma(z_2)$. Equivalently, since the Gaussian convolution is a linear operation, we can express this as the diffusion of $\delta_0$ with $\sigma(z_1)$, followed by diffusion of the resulting profile with the differential diffusion length

$$\tilde{\sigma} = \sqrt{\sigma^2(z_2) - \sigma^2(z_1)}. \tag{2}$$

For the T13 isotope profiles, we account for an approximate average effect of differential diffusion over 2 years by considering the average diffusion lengths calculated over the depth of the T13 profiles before and after downward-advection, neglecting the seasonal variations in firn temperature.

## 2.5 Statistical tests

We use the Kolmogorov–Smirnov (KS) test to assess whether distributions of differences between mean trench profiles vary. Autocorrelation of the data is accounted for with a modified version of the standard test adopting effective degrees of freedom of $n(1-a)$ (Xu, 2013). Here, $n$ is the total number of data points for each profile and $a$ the estimated autocorrelation parameter at lag 1. The KS test compares the empirical cumulative distribution functions of the data and is thus sensitive to differences in both mean and variance.

## 3 Results

### 3.1 New T15 isotope data and qualitative comparison with T13

The two new T15 $\delta^{18}$O trench records measured in 2015/16 (Fig. 3a, b) are qualitatively consistent with the T13 data (Münch et al., 2016) measured 2 years earlier. The isotopic variability within the first metres of firn is characterised by roughly horizontal, alternating layers of enriched and depleted isotopic composition (Fig. 3a, b) which are separated on average by the annual layer thickness of snow (20–30 cm) and thus likely indicative of the climatic seasonal cycle. In addition, stratigraphic noise leads to significant horizontal variability, visible through discontinuous and inhomogeneous layering as well as patchy features, for example at the surface of trench T15–2 (Fig. 3b). We find similar statistical properties for the data of each trench campaign (Table 2). The higher variances in vertical direction of the T15 records are partly expected for autocorrelated data in combination with a larger record length, in addition to the contribution by the strongly enriched layer around 170 cm depth.

Averaging across all individual profiles of each T15 trench reduces the noise level and yields mean profiles that are highly correlated (correlation $r = 0.91$, RMSD $\sim 1.2\,‰$, Fig. 3c) and thus spatially representative. We maximised this match by allowing vertical shifts of the T15–2 mean profile. Using linearly interpolated data on a resolution of $0.5$ cm, we find an optimal shift of $-0.5$ cm. We note the exceptionally high delta values at the top of the T5–2 mean profile which stem from a prominent dune at the trench surface (Fig. 3). However, on the absolute depth scale this top part has no overlap with the T15–1 mean profile and therefore does not contribute to the total T15 mean profile discussed below. Despite their representativity, the T15–1 and T15–2 mean profiles show strong year-to-year variability confirming the discrepancy to local temperature previously found for T13 (Fig. 1). This also becomes apparent through the increase in average T15 summer maxima (Fig. 3c) which is statistically significant ($p < 0.01$) but not captured by the evolution of local summer temperatures (Fig. 1).

Our first findings show that at our study site both the nature of the regional isotope signal and the stratigraphic noise are comparable between the two trench campaigns. In the following sections we quantitatively assess to what extent the original T13 signal can be recovered with the T15 trenches obtained 2 years later. For this task, we use a single data set for T13 and T15 from averaging across each pair of mean profiles (Fig. 4), accounting for the optimal vertical shifts that maximise each inter-trench correlation (Table 2).

### 3.2 Expected isotope profile changes between 2013 and 2015

We analyse to what extent the T13 record can be recovered from the new T15 data, and which changes have modified the original record. Within the 2 years, we expect that the T13 isotope profiles are advected downwards, compressed by densification and smoothed by firn diffusion. Testing for additional isotope modifications hence requires estimating at first the magnitudes of those expected processes. We do this in two ways: Firstly, we use data independent of the trench records. Secondly, to check consistency with the first estimate, we determine the optimal parameter set that minimises the difference between the T13 and T15 mean profiles.

Using the available independent snow stake and density data, we obtain the following estimates: The annual accumulation rates suggest a downward-advection of the T13 profiles after 2 years of $\sim 50$ cm. Further, we expect additional diffusional

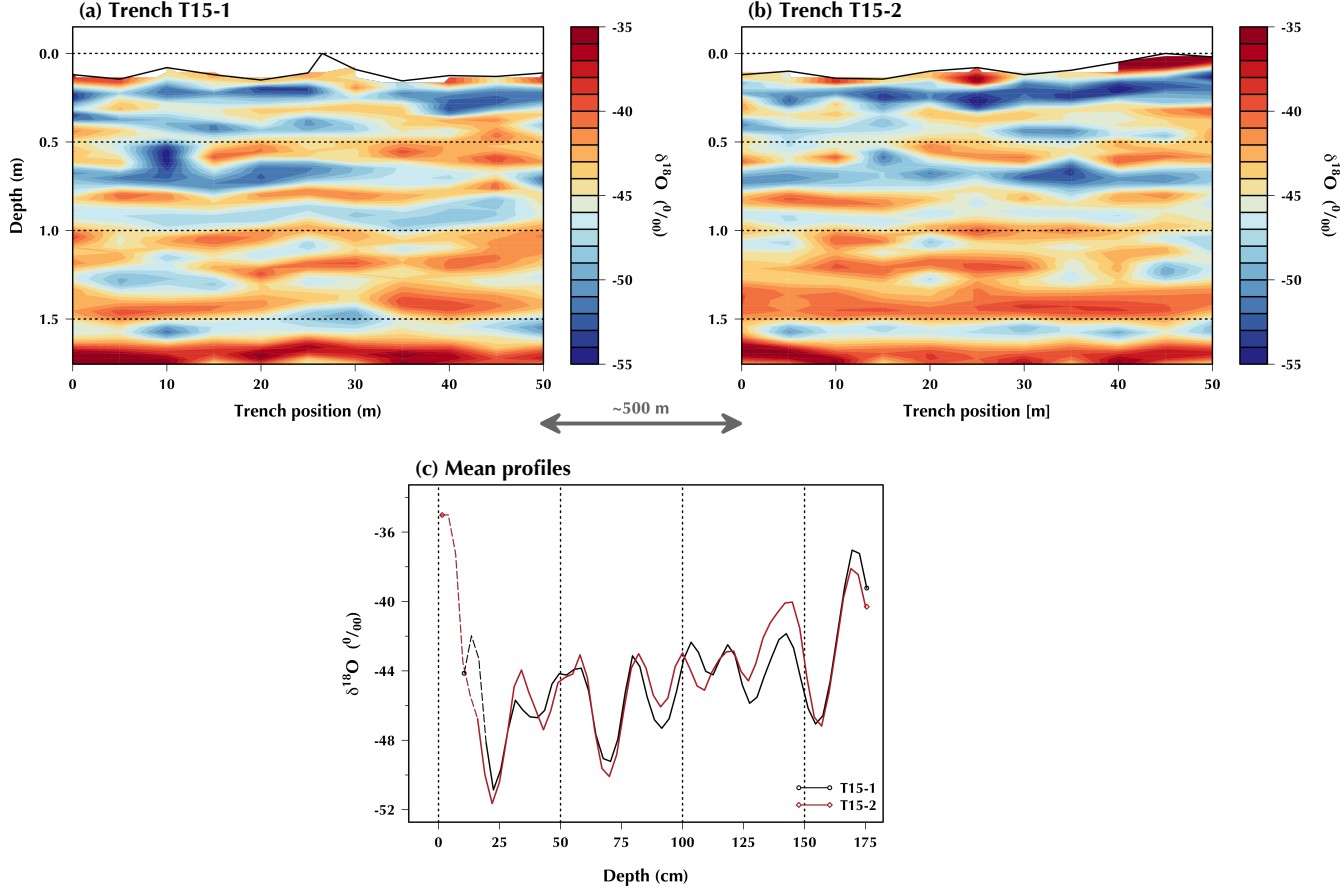

**Figure 3.** The new T15 $\delta^{18}$O data set. Displayed are the isotope records of trench T15–1 (**a**) and trench T15–2 (**b**) as two-dimensional colour images, and the mean profiles from averaging across the individual profiles of each trench (**c**), displayed for the optimal vertical shift of the T15–2 mean profile (see text). The trench surface height profiles are given by solid black lines, the near-surface part of each mean profile is marked by dashed lines since the trench data are incomplete there (see Data and methods). The vertical scale in **a** and **b** is strongly exaggerated.

smoothing of the T13 $\delta^{18}$O profiles according to a differential diffusion length (Eq. 2) of $\tilde{\sigma} \sim 1.9$ cm. The estimated densification rate at the study site of $\sim 2$–$7\% \, \mathrm{m}^{-1}$ implies a compression of the T13 profiles after 2 years of approximately 1–4 cm.

For the second estimate, we vary the three parameters (downward-advection $\Delta$, differential diffusion length $\tilde{\sigma}$, compression $\gamma$) in order to minimise the root-mean square deviation between the T15 and T13 mean profiles. To avoid an influence on our results, we choose the range of tested parameter values independently of the trench data: For the downward-advection, we apply vertical shifts between 40 and 60 cm, comprising the snow-stake based range of the recent annual accumulation rates. We vary the differential diffusion length from 0 to 8 cm, which is equivalent to additional diffusional smoothing of the original T13 mean profile from zero to the maximum possible amount at the firn–ice transition. Finally, compression is applied for values

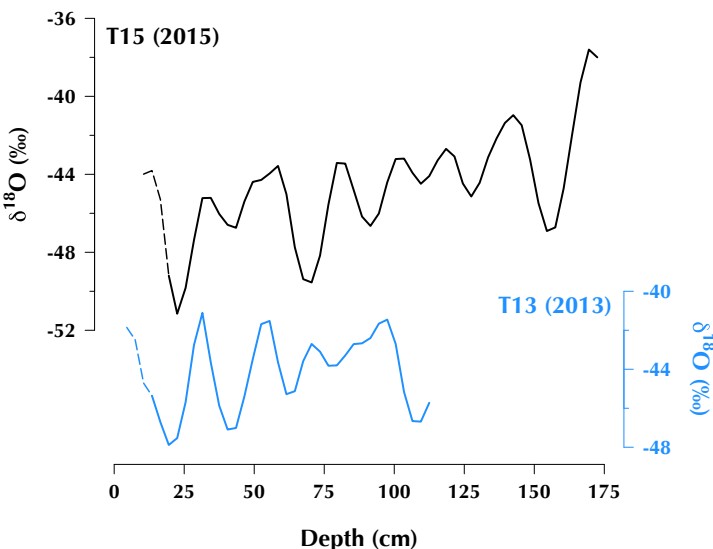

**Figure 4.** The mean oxygen isotope profiles of the T15 (this study) and T13 (Münch et al. (2016)) trenches on their original depth scale. The incomplete surface layer of the trenches is marked by dashed lines.

between $0$ and $10\,\text{cm}$ (equivalent to $0$ to $\sim 5$ times the observed average densification rate). We obtain the best agreement (RMSD $= 0.92\,\text{‰}$, Fig. 5; $r = 0.93$) between the T15 and the modified T13 mean profile ($=$ T13$^*$) for the optimal parameters $\Delta_{\text{opt}} = 50.5\,\text{cm}$, $\tilde{\sigma}_{\text{opt}} = 2.3\,\text{cm}$ and $\gamma_{\text{opt}} = 3.5\,\text{cm}$ (Fig. 6). These trench-based parameter estimates agree reasonably well with the independent estimates from above, showing that the trench data are compatible with our assumptions and parameterisa-

5    tions for downward-advection, densification and diffusion. Indeed, using the independent parameter estimates ($\Delta_{\text{ind}} = 50\,\text{cm}$, $\tilde{\sigma}_{\text{ind}} = 1.9\,\text{cm}$, $\gamma_{\text{ind}} = 2.2\,\text{cm}$ from mean over estimated densification rate) to modify the original T13 mean profile ($=$ T13$^{**}$), results in a deviation from T15 (RMSD $= 0.94\,\text{‰}$, $r = 0.93$) that is only slightly higher compared to T13$^*$.

We note that the largest portion of optimising the fit between T15 and T13$^*$ is accounted for by the downward-advection. This is obvious from only shifting the T13 mean profile vertically to find the maximum correlation with T15, without accounting

10    for diffusion and densification. We find an optimal shift of $48.5\,\text{cm}$ ($r = 0.88$) with a minimum misfit of RMSD $= 1.07\,\text{‰}$ (Fig. 5). Thus, the gain in RMSD is only small when adding diffusion and densification according to T13$^*$ (black dot in Fig. 5) or T13$^{**}$, but still appears significant given the above found consistency in magnitude of the trench-based and independent estimates. This is further supported by the fact that no second minimum in RMSD exists outside the region bounded by the contour line of only downward-advection (RMSD $= 1.07\,\text{‰}$, Fig. 5) where the magnitudes of diffusion and densification are

15    unrealistically high.

The visual agreement of the trench mean profiles after modifying T13 according to downward-advection, diffusion and densification is remarkable regarding cyclicity and, to a lesser extent, the amplitude of the isotope variations (Fig. 6b). However, deviations occur throughout most of the record's overlap (Fig. 6b) and are even amplified there where the amplitude of the T13

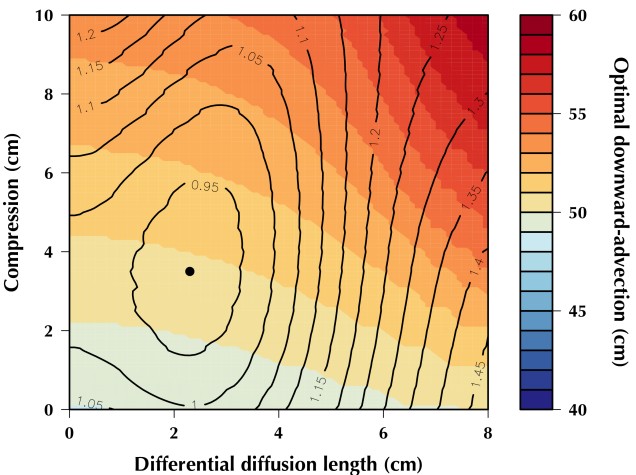

**Figure 5.** Effect of downward-advection, firn diffusion and linear compression due to densification on the misfit (root-mean square deviation, RMSD) between the T15 and the modified T13 mean profile. We record the RMSD for each point in the three-dimensional parameter space of downward-advection, compression and diffusion. For each diffusion–compression pair, the figure shows the local minimum in RMSD (contour lines) from varying accross the range of advection values, hence the RMSD for the optimal downward-advection (colour scale). The global minimum in RMSD is marked with a black dot. Varying the downward-advection has in fact the largest influence on the RMSD.

profile prior to diffusion was smaller than for T15 (depths of $\sim 70\,\mathrm{cm}$ and around $\sim 125\text{–}140\,\mathrm{cm}$). Here, locally additional diffusion does not lead to an improved match, although overall it reduces the mismatch between the profiles (Fig. 5). In general, the profile deviations are relatively large compared to the influence of firn diffusion and densification on the original T13 profile (Fig. 6a), which calls for studying further processes in order to explain them.

### 3.3 Do the remaining differences represent temporal or spatial variability?

We have shown that downward-advection, firn diffusion and densification contribute to the temporal modification of the original T13 profiles as expected from independent data and theoretical considerations. Taking these processes into account leads to a good match of the trench mean profiles (Fig. 6b). Most of the match is achieved by accounting for the downward-advection; adding the effects of diffusion and densification yields a slightly further improvevement (gain in RMSD of $\sim 0.15\,‰$). However, still deviations between the profiles on the order of $\sim 1\,‰$ RMSD remain. These can have two causes: firstly, additional temporal changes driven by unaccounted post-depositional processes such as firn ventilation or sublimation; secondly, remaining spatial variability since we average a large but finite number of records which do not originate from the exact same position. We can thus deduce the importance of additional post-depositional change for our study site if we quantify the contribution of spatial variability. In the following, this is done in two ways: (1) by using the statistical model for stratigraphic noise, and (2) by analysing the distributions of the profile differences.

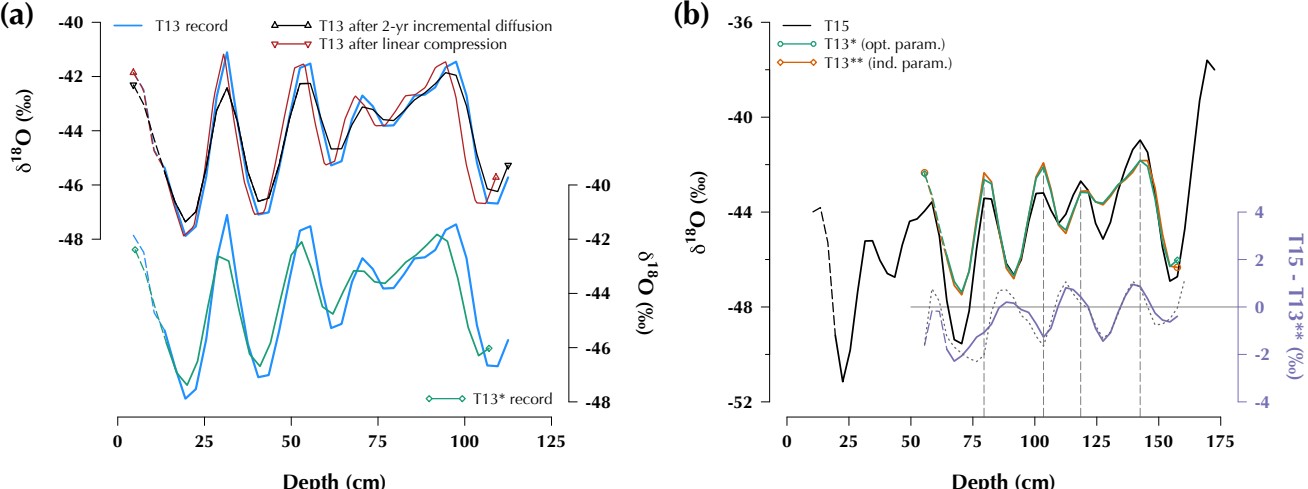

**Figure 6.** Expected changes of the T13 and comparison to the T15 mean profile. **a**: (Upper panel) The original T13 mean profile (blue) and its modification by diffusion (black: 2-year diffusion with differential diffusion length $\tilde{\sigma} = 2.3\,\mathrm{cm}$) as well as densification (red: linear compression of $\gamma = 3.5\,\mathrm{cm}$). (Lower panel) The original T13 mean profile (blue) compared to the joint effect of 2-year diffusion and linear compression (green, T13*). **b**: The T15 mean profile (black) in comparison to the T13 mean profile after modifying the latter according to (1) the optimal parameters for downward-advection, incremental diffusion and linear densification (green, T13*), and to (2) the corresponding parameters obtained independently from the trench records (orange, T13**). Additionally, the difference between T15 and T13** is shown (violet lines, axis to the right). For comparison, the grey dotted line marks the difference between T15 and T13 only shifted optimally ($\Delta = 48.5\,\mathrm{cm}$). Vertical dashed lines indicate the isotopic summer maxima which are not in phase with the difference curve.

According to the statistical noise model, the effective number of profiles that contribute to the T13 and T15 mean profiles (Eq. 1) is $N_{\mathrm{eff}} = 13$ for T13 and $N_{\mathrm{eff}} = 20$ for T15. The residual noise of the mean profiles arising from spatial variability is thus the noise level before averaging ($\omega \sim 5\text{--}7\,(67\,\%\,\mathrm{CI}\!:\!4\text{--}8)\,(\text{‰})^2$, Table 2) divided by 13 and 20, respectively. We assume that the residual noise terms are independent of each other. Therefore, the uncertainty of the difference between the T13 and T15 mean

5    profiles due to stratigraphic noise is the sum of each residual spatial variability, or $\sim 0.6\text{--}0.9\,(0.5\text{--}1.0)\,(\text{‰})^2$. For comparison, the square of the RMSD between the T13* (T13**) and T15 mean profile (the "temporal varaibility") is $0.85\,(0.88)\,(\text{‰})^2$. This agreement between RMSD and estimated residual spatial variability indicates that the remaining profile differences between the modified T13 mean profile and T15 (Fig. 6b) might be likely consistent with stratigraphic noise. We note however that the squared RMSD lies towards the upper end of the estimated range of residual stratigraphic noise. This also applies to the RMSD

10   between the T15–1 and T15–2 mean profiles (square of RMSD of $1.44\,(\text{‰})^2$ vs. uncertainty from residual stratigraphic noise of $\sim 1.0\text{--}1.4\,(0.8\text{--}1.6)\,(\text{‰})^2$). This could indicate that part of the spatial variability on the scale of the inter-trench distances ($\sim 500\,\mathrm{m}$) is not explained by our stratigraphic noise model.

We therefore make a formal statistical test comparing spatial and temporal variability, which accounts for the full extent of spatial uncertainty and makes no assumption about the covariance of the noise, by analysing the deviations between the

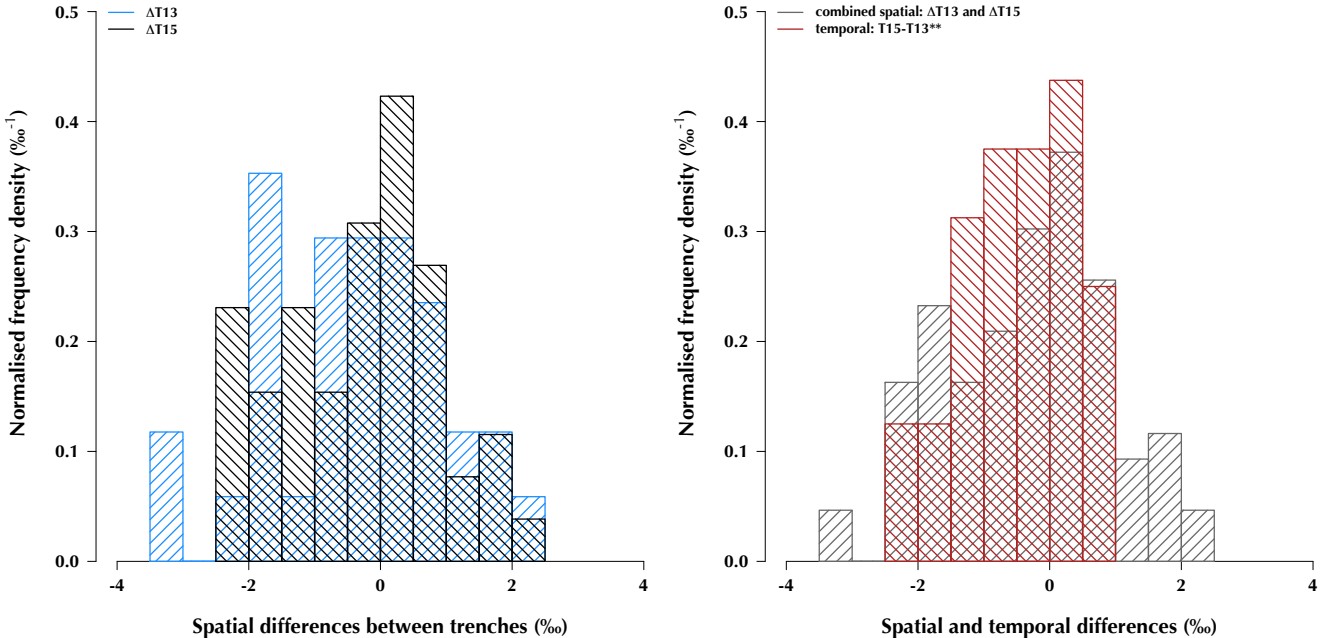

**Figure 7.** Variability of the trench data sets. The histograms depict (**a**) the distribution of the spatial differences between the two mean profiles of the T13 (T13–1 vs. T13–2, blue) and the T15 trenches (T15–1 vs. T15–2, black), and (**b**) the combined distribution from **a** (grey) compared to the distribution of the temporal differences between the T15 and the T13** mean profiles (red). All distributions' mean values are not significantly different from zero (all $p \geq 0.4$, accounting for autocorrelation).

mean trench profiles. We find that the spatial differences between the mean profiles of each trench campaign (T13–1 vs. T13–2 and T15–1 vs. T15–2, Fig. 7a) are statistically indistinguishable ($p > 0.5$ from modified KS test, combining all possible directions of calculating the differences), meaning that the statistical distributions of the inter-trench scale spatial variability are not significantly different between the years 2013 and 2015. This suggests that we can combine these spatial differences

5   as a joint measure of spatial variability and compare them to the temporal differences between the T15 and the modified T13 mean profile (Fig. 7b). Applying the modified KS test once more, also here the null hypothesis that both differences follow the same distribution cannot be rejected (all $p > 0.5$ for using T13** to avoid overfitting). Thus, the temporal differences between T13** and T15 likely just arise from the fact that the trenches have different locations and can be therefore explained by spatial variability alone.

10      In summary, both methods show no evidence for any temporal changes of the trench record over the course of 2 years apart from downward-advection accompanied by firn diffusion and densification. The remaining deviations that are observed between the mean profiles of the two trench campaigns can be entirely explained by residual spatial variability arising from stratigraphic noise.

## 4    Discussion

We presented and analysed a new extensive data set of 22 oxygen isotope profiles obtained at Kohnen Station from two $50\,\mathrm{m}$ long and $\sim 180\,\mathrm{cm}$ deep snow trenches. The new trench campaign was designed such that it allows for a direct comparison with a trench data set obtained from the same site 2 years earlier in order to test for post-depositional effects. In the following, we first discuss our results concerning the expected processes that have influenced the trench isotope profiles over the observed time period, then our findings regarding the possibility of additional post-depositional changes.

### 4.1    Densification, diffusion and stratigraphic noise

We found a strong resemblance between the mean oxygen isotope profiles from the trench field campaigns of 2013 and 2015 (Fig. 6b), achieved mostly by accounting for the downward-advection and further improved by adding the effects of water vapour diffusion within the firn and firn densification that occurred during the 2 years (Fig. 5). The estimated magnitudes of these processes obtained from matching both records are consistent with independent estimates from snow stakes, diffusion theory and independent density profiles.

The estimated small compression of the T13 profiles is reasonable given the low densification rate observed in the top metres of nearby firn cores. However, our assumption of a linear profile compression with depth is certainly a rough approximation given the actually observed seasonal firn density variation (Laepple et al., 2016), which might indicate a stronger density change with depth of summer compared to winter layers. However, in general the seasonality of densification in Antarctic firn is largely unclear (Laepple et al., 2016, and references therein).

Our data-based estimate of differential firn diffusion agrees with theoretical expectations and in total leads to a further reduction in RMSD between the T13 and T15 mean profiles compared to the case of downward-advection and densification alone (Fig. 5). In detail, the diffusion correction improves the match of the trench mean profiles in the medium depth range but also results in higher deviations of the profile minima at the top and bottom part of the overlap (Fig. 6), where the amplitude of the T13 profile had been smaller than for T15 already prior to diffusion. Part of this mismatch might be reduced by accounting for the seasonally varying firn temperature resulting in stronger (weaker) attenuation of summer (winter) layers caused by the seasonal difference in diffusion length which is largest close to the surface (Simonsen et al., 2011). In general, firn diffusion is still an active area of research (van der Wel et al., 2015), and progress in this field could conceivably result in an improved understanding of our data.

Stratigraphic noise is a major contribution to the overall variability of isotope profiles (Fisher et al., 1985; Karlöf et al., 2006; Münch et al., 2016). Our large trench data set allows a significant reduction of the noise level by averaging across the single profiles. This is done in two steps: First, we average across the local (intra-trench) scale; then, we average the resulting mean profiles to account for potential uncertainties on the $500\,\mathrm{m}$ (inter-trench) scale. Furthermore, we can estimate the remaining uncertainty of the trench mean profiles based on our theoretical understanding of stratigraphic noise. As a result, we found that the difference of the T13 and T15 mean profiles still exhibits an uncertainty of $\sim 0.77\text{--}0.95\,\text{\textperthousand}$ (SD). Thus, the trench data allow us to detect any additional post-depositional changes of the T13 profiles that exceed a detection limit of $\sim 1\,\text{\textperthousand}$ RMSD.

Obviously, a lower detection limit would be beneficial but is in practice constrained by the amount of field work, given the high local stratigraphic noise level as observed from the mean horizontal isotope variability (Table 2).

## 4.2 Additional post-depositional modifications

Based on the above results we have shown that the remaining differences between the 2013 and 2015 data sets are, after accounting for downward-advection, firn diffusion and densification, likely consistent with spatial variability from stratigraphic noise. In other words, we conclude that at our study site the impact of any additional post-depositional changes of the isotopic composition of the firn, below $\sim 10\,\mathrm{cm}$, is on average below the residual stratigraphic noise level, thus $\ll 1\,\text{‰}$ RMSD. We limited our conclusion to this depth range due to the applied absolute depth scale resulting in a lower and varying number of available data points in the surface layer. However, looking at this part of the modified T13 mean profile (dashed lines of T13$^*$ or T13$^{**}$, Fig. 6b) also does not show any solid evidence for additional post-depositional changes.

Our conclusion is also supported by comparing the nature of the differences between the mean profiles (Fig. 6b) with the expected effect of post-depositional modification processes. Studied processes all point to isotopic enrichment, such as sublimation (Stichler et al., 2001; Sokratov and Golubev, 2009) and wind-driven firn ventilation (Town et al., 2008). Specifically, the latter modelling study showed that firn ventilation can result in isotopic annual-mean enrichment from the strong enrichment of isotopic winter layers, compensating an observed slight depletion of summer layers. For South Pole conditions (annual-mean temperature $-50\,°\mathrm{C}$, accumulation rate $84\,\mathrm{mm}$ w.eq. $\mathrm{yr}^{-1}$, mean surface wind speed $5\,\mathrm{m\,s}^{-1}$), the effect amounts to $\sim 3\,\text{‰}$ for firn ventilation until the layers are advected below the influence of the atmosphere, thus after several years (Town et al., 2008). The environmental conditions at the South Pole are comparable to Kohnen Station (Table 1), suggesting a similar influence of ventilation on the isotopic composition of the firn. The higher temperatures at Kohnen Station would even imply a slightly stronger enrichment (Town et al., 2008). However, if we analyse the difference curve of the T15 and T13$^{**}$ mean profiles (Fig. 6b) we find no evidenve for firn ventilation. Comparing the direct seasonal counterparts, the first winter layer, which was closest to the surface at the time of excavation of T13 and thus presumably being under strongest influence of the atmosphere, is more depleted in isotopic composition in T15 than in T13$^{**}$, in contrast to the expectation from firn ventilation. Moreover, despite the fact that the first three summer layers exhibit more depleted values, which would be in line with ventilation, the remaining summer layers do not confirm this finding, and none of the average annual differences show enrichment: for the first annual cycle, T15 exhibits an average difference from T13$^{**}$ of $-1.6\,\text{‰}$ ($-1.3\,\text{‰}$ including the surface region), for the other annual cycles the averages are $-0.4$, $\pm 0$ and $-0.1\,\text{‰}$. Also in general, the difference curve (Fig. 6b) does not show any clear seasonal timing which might be expected for a systematic post-depositional modification. Instead, minimum and maximum differences appear rather randomly across the seasons. In addition, the global average difference of about $-0.45\,\text{‰}$ is not significantly different from zero ($p = 0.4$, accounting for autocorrelation). We nevertheless note that the RMSD of the first overlapping annual cycle is above our stated detection limit for post-depositional change. However, this limit applies to the average over the record's entire overlap and does not account for the possibility of autocorrelated differences. Finally, we note the seeming increase with depth of the annual-mean differences towards more positive values (Fig. 6b), which is also indicated by the slight skewness of the corresponding histogram (Fig. 7b). However, the trend is not strongly significant ($p = 0.12$,

accounting for autocorrelation), and the KS test of the distribution of the differences showed that mean and variability of the residual temporal differences are likely explained by the spatial distribution alone. In addition, we obtain similar results (not shown) when we apply our analysis to the trench d-excess ($d := \delta D - 8 \cdot \delta^{18}O$) data, a second-order parameter potentially more sensitive to post-depositional fractionation processes (Touzeau et al., 2016). The spatial and residual temporal differences between the corresponding d-excess mean profiles follow the same distribution ($p > 0.5$), and the histogram of the temporal differences is even more symmetric than for $\delta^{18}O$.

In summary, all evidence suggests that post-depositional modifications from firn ventilation, or sublimation, are unlikely to contribute to the deviations between the T15 and the modified T13 mean profiles, and that the shape of the difference curve only arises from the statistical nature of stratigraphic noise, smoothed by diffusion. We nevertheless note the possibility that additional post-depositional changes are still present at Kohnen Station but not detectable in our analysis. Wind-driven firn ventilation might exist but its effect being much weaker than expected and thus masked by the stratigraphic noise level. One possible explanation for the discrepancy between the firn ventilation model results and our data could be that the model misrepresents the isotopic signature of the surface vapour advected into the firn. Another possibility are weaker firn temperature gradients at Kohnen Station compared to the South Pole, preventing significant vapour deposition. Assessing these possibilities in detail is however beyond the scope of our study. Interestingly, if the seeming trend in difference values were significant, it would suggest an oriented post-deposition process that is yet unknown. In any case, the stronger profile differences for the first overlapping annual cycle might indicate modification processes that are constrained to the very surface layer. In addition, the RMSD between T15 and T13* can be further minimised if one allows shifts in the mean value of the T13 profile (new minimum RMSD of $-0.82\,‰$ for a shift in mean of $-0.4\,‰$) which is an interesting observation, yet without any obvious explanation. However, based on the presented evidence, these possibilities are speculative and further field studies are needed to test them.

Our study underlined the pronounced discrepancy at Kohnen Station between inter-annual variations of isotope ratios in the firn and local temperatures and showed that this feature is not only spatially (over distances of $\sim 500\,\mathrm{m}$) but also temporally representative over a period of 2 years. Furthermore, given the sum of our above findings, it is unlikely that post-depositional modifications of the isotopic composition of the open-porous firn (below depths of $\sim 10\,\mathrm{cm}$, and probably also not in shallower depths) are the cause of the observed discrepancy. Since a strong relationship between isotopes in precipitation samples and local temperature has been observed at different sites of the East Antarctic Plateau (Fujita and Abe, 2006; Touzeau et al., 2016), this cause must instead be sought in processes working directly at or above the firn surface. At least two explanations for this seem possible. (1) Seasonal variation and intermittency of precipitation cause the discrepancy between isotope and local temperature data (Sime et al., 2009, 2011; Persson et al., 2011; Laepple et al., 2011). At Kohnen Station, a large part of the annual accumulation is assumed to occur in winter since little or no precipitation is observed in the summer field seasons. However, the exact seasonal and inter-annual variation of accumulation is still unclear due to the lack of sufficiently precise, year-round observations (Helsen et al., 2005). The available surface height changes derived from sonic altimeters of automatic weather stations are difficult to separate into events of drifting snow and true snowfall (Reijmer and van den Broeke, 2003). (2) Isotope modification occurring directly at or above the surface is the key driver for shaping the inter-annual isotope variations.

Such processes might be acting on falling, loose or drifting snow, or on the top layer (first few centimetres) of deposited snow (Ritter et al., 2016; Casado et al., 2016). The fact that our trench records are reproducible on spatial scales of at least $500\,\mathrm{m}$ implies that the atmospheric parameters and conditions controlling potential processes would also need to be spatially coherent.

## 5 Conclusions

Many studies, including our present one, show that inter-annual isotope records from the dry East Antarctic Plateau are inconsistent with local temperature variations. However, beyond simply stating the problem, we take two steps further: (1) We use the average over $2 \times 11$ isotope profiles to obtain a spatially representative record. (2) We designed our study such that it allows testing for post-depositional effects over a time span of 2 years.

Our results provide important constraints on the formation of the stable water isotope signal and its propagation with depth in East Antarctic firn: The trench records show a pure downward-advection of the isotope signal within the open-porous firn ($\gtrsim 10\,\mathrm{cm}$ depth), further influenced only by firn diffusion and densification, with no evidence for substantial additional post-depositional modification. Hence, once the signal is archived at this stage, we do not expect any significant change of the mean values deeper down, reinforcing the credibility of palaeoclimate studies using ice core isotope data. However, from our analyis we can constrain post-depositional changes only down to the level of stratigraphic noise. Therefore, qualitatively, firn ventilation and sublimation might still be present but their effect being very small, or constrained merely to the surface layer for which the lower number of data points in our study prevents quantitative analyses. These constraints lead us to conclude that the observed discrepant isotope–temperature relationship on the inter-annual timescale must be caused either by processes prior to or during deposition.

To improve our understanding of the inter-annual isotope signal, we suggest a mixture of field and modelling efforts. Year-long isotope studies (e.g. in seasonal intervals) with a focus on the near-surface would help to constrain isotope modifications at the interface of surface snow and atmosphere. Further, the role of precipitation and accumulation intermittency has to be clarified, e.g. through measuring wet-depositioned tracers and improved accumulation measurements. These studies should optimally be accompanied by monitoring and modelling of the atmospheric water vapour isotopic composition as well as modelling of the potential exchange and fractionation processes between the loose or deposited snow at the surface and the overlying atmosphere.

Our results again underline the role of stratigraphic noise for the total variability of isotope records. Spatial averaging is thus essential to improve the signal-to-noise ratio and thereby to separate spatial from temporal variability. Alternatively, single records can only be compared faithfully for temporal changes when their spatial separation is well below the spatial decorrelation length of the stratigraphic noise, which minimises the amount of spatial variability between the records. The effects of potential isotope modifications depend substantially on the time the surface is exposed to the atmosphere, thus on accumulation rate and seasonal timing of precipitation. Comparable recovering efforts at other ice-coring sites are hence highly needed. Our data indicate that present models might overestimate the expected influence of wind-driven firn ventilation; however, regions with higher wind speeds and lower accumulation rates might still be susceptible towards post-depositional

changes within the open-porous firn. A deeper understanding of the isotope signal formation in Antarctic firn is, beyond holding intrinsic interest, essential to decipher the temperature signal archived in ice core records and thus crucial for their palaeoclimatic interpretation.

*Data availability.* The trench stable water isotopologue data presented in this study are archived at the PANGAEA database (http://www.pangaea.de) under doi (link will follow). PANGAEA is hosted by the Alfred Wegener Institute Helmholtz Centre for Polar and Marine Research (AWI), Bremerhaven, and the Center for Marine Environmental Sciences (MARUM), Bremen, Germany.

*Author contributions.* Thomas Münch, Thomas Laepple, Sepp Kipfstuhl and Johannes Freitag designed the trench campaign, Sepp Kipfstuhl and Thomas Münch led the field work. Thomas Münch and Thomas Laepple designed the analysis. Thomas Münch performed the research and wrote the manuscript. Hanno Meyer supervised the isotope measurements. All authors contributed significantly to the discussion of the results and the refinement of the manuscript.

*Competing interests.* The authors declare that they have no conflict of interest.

*Acknowledgements.* We would like to thank all scientists, technicians and the logistic support who worked at Kohnen Station in the 2014/2015 austral summer, especially Holger Schubert and Tobias Binder, for assistance in creating the trench data set. Thanks are also due to Christoph Manthey of the isotope laboratory in Potsdam for assistance in measuring the isotope data. We are grateful to Gerit Birnbaum and Carleen Tijm-Reijmer for providing the AWS9 data. This work significantly benefited from discussions with Mathieu Casado and from proofreading and many valuable comments by Andrew Dolman. All plots and numerical calculations were carried out using the software R: A Language and Environment for Statistical Computing. We acknowledge support of this work by the Initiative and Networking Fund of the Helmholtz Association Grant VG-NH900.

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
