# Peer review of "Constraints on post-depositional isotope modifications in East Antarctic firn from analysing temporal changes of isotope profiles"

_The Cryosphere, 2017_

## Referee Comment (RC1) · Anonymous Referee #1 · 4 May 2017

______________________________General comments______________________________

This article presents new measures of isotopic compositions (d18O, d-excess) in the first 2 meters of snow at Kohnen (Antarctica). These measurements are used to evaluate how the isotopic signal is modified with time (over a two-year interval), after deposition, at this site. The authors also present a simple model including 3 post-deposition processes, and use it to simulate the evolution of d18O values for the same period of time. The model and data results are coherent with each other. The authors conclude that no other processes (besides these three) are necessary to account for d18O evolution in the snow layers. Besides this study of post-deposition, the authors compare the spatially averaged d18O profile in the snow to measured temperature evolution

(AWS) and note a strong discrepancy. Since post-deposition processes do not explain this discrepancy, they propose that processes before or during deposition have to be investigated.

I recommend that this paper be accepted with moderate revisions.

1) The data presented here are crucially needed at the moment. They not only represent a huge amount of field work and analysis, but also respect a carefully designed set-up to ensure the quality of the signal retrieved by minimization of horizontal noise. Such high-quality data are exactly what is required to evaluate quantitatively the impact of post-deposition processes.

2) The quantitative evaluation of the three processes studied through minimization of RMSD is clear, and the magnitudes obtained are coherent with independent estimates.

3) However, the articulation between the strategy of the field experiment and the broader issue of the discrepancy between interannual temperatures and interannual d18O could be more detailed in Introduction.

4) The authors could nuance their conclusion that post-deposition processes are unable to produce the interannual variability of d18O observed. Only three processes have been evaluated quantitatively, the others are rejected based only on qualitative observations (and are still subject of research).

/

/

/

/

/

/

/

_____________________________Specific comments___________________

/

/

ABSTRACT

O______'Here we reject the hypothesis of post-depositional change within the open-porous firn beyond diffusion and densification.' This sentence is unclear. Is it possible to use affirmative form?

O______'These results show that the discrepancy between local temperatures and isotopes most likely originates from spatially coherent processes prior to or during deposition, such as precipitation intermittency or systematic isotope modifications acting on drifting or loose surface snow.' Why did you choose to evaluate post-deposition processes and not precipitation intermittency in this study? The latter is a strong candidate for the observed discrepancy. Is it due to a lack a measurements?

/

/

/

INTRODUCTION

O______When you say that diffusion and condensation 'only smooth and compress the original signal', you should precise that you are talking about vapor diffusion against isotopic gradients.

O______ 'In contrast, the low local annual accumulation rates and potential seasonal intermittency of precipitation increase the time the surface is exposed to the atmosphere (Town et al., 2008; Hoshina et al., 2014) and therefore to processes that might alter

the snow's original isotopic composition.' The intermittency of precipitation does not only favor post-deposition processes through exposition to the atmosphere; it can also shape the d18O signal because of irregular accumulation.

O______'These processes can act either on loose snow in the post-condensational phase (falling or drifting snow), . . .' Could you precise which processes are active then? It is not wind redistribution, since these processes have to be spatially coherent.

O______'This discrepancy stresses the importance of contributions other than regional temperature alone to the formation of the isotope signal. /// In this study, we investigate whether post-depositional isotope modifications in the open-porous firn contribute to the observed discrepancy between isotopes and local temperature at Kohnen Station.' This transition is very short. Could you indicate briefly what are the other contributions and why this study is dedicated to post-deposition?

Figure 1.

O______Do you have information on precipitation amounts over this period? Or on summer d18O in the snowfall? Does the summer d18O in the snowfall follow the evolution of summer temperatures? If precipitation amounts are unknown, please state it here, not later in the Discussion. . . It will be easier to understand why you focus on post-deposition processes.

O______'. . .we have designed our study such that it allows for the first time to quantitatively follow the isotopic changes and thus to test for post-depositional effects over a time span of 2 years.' What do you expect for the evolution of the variability over 2 years? An attenuation or an amplification? If you expect only an attenuation, then post-deposition is obviously not responsible for the discrepancy between temperature and d18O interannual variations (attenuating a flat profile will not lead to increased variability). If you expect amplification, then why do you simulate only 'attenuating' post-deposition processes?

/

/

/

RESULTS

Table 2.

O______ 'The higher variances in vertical direction of the T15 records are partly expected for autocorrelated data in combination with a larger record length,' It seems also stronger for the horizontal variability. Do you have an explanation for that? There is also a strong increase of the signal-to-noise ratio. Does it mean that the mean profile in 2013 is less well known?

Figure 3.

O______Considering only the part of the profiles that is complete, there seems to be an increase of d18O with depth. The shallowest winter (24 cm) has a very low value compared to the deepest winter (153 cm). There is a similar trend for summers (-37‰ for the summer at 173 cm and -44‰ for the summer at 33 cm). Is it possible to test this trend with a linear regression? Do you have information on the continuation of this trend at greater depths? If this trend is verified, what process could be responsible of such an increase?

Figure 4.

O______It is really difficult to compare quantitatively the two curves on this figure, because they are not superposed. Could you put them on the same d18O scale, and shift the 2013 curve 'optimally'?

«<Figure 4: superposed»> see attached figure (Figure 1)

O______ 'In the 2 years, the T13 isotope profiles are advected downwards, compressed

by densification and smoothed by firn diffusion.' Attenuation is not very clear here. There is attenuation between 75 and 120 cm depth (blue zone). However, between 60 and 75 cm depth and also between 125 and 150 cm depth the profile after two years (2015) has larger amplitude (red zones). Adding attenuation to the initial d18O profile from T13 would increase the agreement in the blue zone, but decrease the agreement in the red zones.

Figure 5

O______ 'For the downward-advection, we apply vertical shifts between $\Delta$ = 40 and 60 cm,' This range is too large to stay within the bounds of the first winter minimum (47-53 cm would be enough) and too small to permit the shifting of the curve by one cycle (shift of 25-75 cm required). How is it possible that 60 cm become an optimum (it should lead to anti-correlation)?

O______ Compression higher than 6 cm or diffusion length higher 4 cm leads to RMSD higher than 'doing nothing' (1.05 at the point of origin). This is interesting as it gives an upper bound for the impact of these processes. It also confirms the estimates from independent datasets.

Figure 6

O______ 'We obtain the best agreement (RMSD = 0.92h, Fig. 5; r = 0.93) between the T15 and the modified T13 mean profile (= T13*) for the optimal parameters $\Delta$opt = 50.5cm, $\sigma$opt = 2.3cm and $\gamma$opt = 3.5cm (Fig. 6).' Even if adding attenuation generally increases agreement with 2015, is it really the best scenario to apply here (considering red zones)? If the diffusion length was computed only on the zone where attenuation is evident (between 75 and 120 cm) would it have the same value?

O______ Did you try to move the profile of T13 vertically (more or less enriched in heavy isotopes) to get a better fit? Of course the processes tested here would not lead to a change in the mean value, but it could give information on other processes (maybe for

discussion).

O_____ Could you give us an estimation of the attenuation due to diffusion? It could be useful for future comparisons (to other data or models). Roughly from the graph (T13*), the half-attenuation seems to be of ∼0.6 ‰ and the initial half-amplitude of about 2.2 ‰ which would correspond to a quite strong attenuation, of the order of 25 % over two years. What would be the attenuation in the 'blue zone': 75-120 cm depth?

O_____ p10 l20: 'can be seen' and l21: 'clearly': It would be easier to see the improvement if there were somewhere a figure showing T13 (unmodified) and T15 superposed. Without this figure, the term 'seen' should be avoided/replaced.

O_____ 'Nevertheless, both processes play a significant role in explaining part of the temporal changes. This can be seen if we only shift the T13 mean profile vertically to find the maximum correlation with T15...' Is the RMSD of 'only compression' different from the one of T13*? How much improvement is obtained by adding the diffusion to the 'compression only' experiment?

O_____ 'deviations especially remain around the isotopic extreme values, in particular for the first overlapping cycle and the depths around 100 and 125–140 cm.' As expected, the deviation after post-deposition is high mostly in the red zones (first cycle, 125-140 cm), where the amplitude in 2015 is larger than the amplitude in 2013. For these zones adding diffusion leads to higher deviations than doing nothing (and the term 'remaining difference' is maybe not the best adapted).

O_____ What do you call 'extreme values'? All the extremums? Or only the summer at 175 cm and the winter at 70 cm? If you are talking about the extremums, then there is a contradiction with p. 14: 'Furthermore, the difference curve (Fig. 6b) does not show any clear seasonal timing...'

O_____ 'This gives a best shift of 48.5 cm, but clearly the agreement is less pronounced (RMSD = 1.1‰ r = 0.88) compared to...' On the Figure 5, at the point of

origin (no diffusion, no densification), the RMSD is 1.05. It reaches 0.92 ‰ for optimal compression and diffusion. Thus these two processes are a step in the right direction, but finally do not improve the RMSD very much.

O______ p. 10, l25: 'Taking these processes into account leads to a good match of the trench mean profiles (Fig. 6b). However, deviations on the order of 0.9–1‰ remain.' What were the deviations before taking them into account?

O______ 'These can have two causes: firstly, additional temporal changes driven by un-accounted post-depositional processes;' Could you precise what other processes you are thinking about? Or maybe just make a note toward the section where these unac-counted processes are discussed? Listing possible processes could help to research specific features expected in the remaining variability.

O______ 'secondly, remaining spatial variability since we average a large but finite num-ber of records which do not originate from the exact same position.' It seems coherent to evaluate the remaining variability as spatial noise, if this variability is random. How-ever, it may not be the case here (slight trend toward higher values with depth, see below).

O______ 'The agreement of both estimates indicates that the remaining profile differ-ences between the modified T13 mean profile and T15 (Fig. 6b) can be entirely ex-plained by spatial variability through stratigraphic noise. We note however that the squared RMSD lies at the upper end' If there is still a doubt in your mind after the mathematical demonstration, why do you use the term 'entirely' in the first sentence? This term also seems in contradiction with the end of the paragraph. To facilitate read-ing, you could add a layer of uncertainty such as: 'At first order, the agreement of both estimates indicates'

Figure 7

O______ 'We find that the distributions of the spatial differences between the mean

profiles of each trench campaign (T13–1 vs. T13–2 and T15–1 vs. T15–2, Fig. 7a) are statistically indistinguishable' Could you explicit the results of these tests with simple words? What is the more general conclusion of this first test? That the sampling strategy has no influence on the results? That the uncertainty is the same for T13 and T15?

O\_\_\_\_\_ 'More importantly, the combined distribution of spatial variability is also indistinguishable from the distribution of the temporal differences between the T15 and the modified T13 mean profile' Does this test evaluate if the difference between T13** and T15 is more than just the difference between T13** and T15 that comes from having a different location?

O\_\_\_\_\_ How do you 'combine' spatial differences between trenches? The distances considered are not exactly the same (∼350 m between T13-1 and T13-2; ∼500 meters between T15-1 and T15-2; ∼200 m between the mean T13 position and the mean T15 position). Do you apply a weighting by distance?

/

/

/

DISCUSSION

\_\_\_\_\_\_\_\_Densification, diffusion and stratigraphic noise\_\_\_\_\_\_\_\_

O\_\_\_\_\_ 'We found a strong resemblance...' This 'strong resemblance' is largely brought by moving downward the profile (advection). The impact of compression and diffusion, even if it is significant, is still very small.

O\_\_\_\_\_ 'our assumption of a linear profile compression with depth is certainly a rough approximation given the actually observed seasonal firn density variation (Laepple et al., 2016).' In what direction would that process intervene? Preferential compression

of summers or winters?

O_____ 'In detail, the diffusion correction improves the match of the trench mean profiles in the medium depth range but also results in higher deviations of the profile minima at the top and bottom part of the overlap (Fig. 6).' This observation is much welcome but should have come earlier in the manuscript, when the deviations are first described.

O_____ 'Part of this mismatch might be reduced by accounting for the seasonally varying firn temperature resulting in stronger (weaker) diffusion for summer (winter) seasons (Simonsen et al., 2011).' How exactly? Does this mean that summers would be more attenuated than winters (due to stronger attenuation when they are still at the surface)? What about temperature gradients? They might not only favor attenuation, but also redistribute heavy and light isotopes vertically.

________________Additional post-depositional modifications________

O_____ '...any additional post-depositional changes of the isotopic composition of the firn, below 10 cm, must be on average clearly below the residual stratigraphic noise level, thus « 1‰' Thus the change can be of more than 1 ‰ as long as it goes on opposite directions at top and at bottom (the average being zero)?

O_____ 'This conclusion is also supported by comparing the qualitative nature of the differences between the mean profiles (Fig. 6b)' Regarding this difference (violet curve): is it possible to add the zero line, to discriminate between positive and negative differences? Is it possible to add the difference T15-T13 (with optimum downward advection), to see where the post-deposition has been most effective?

O_____ 'the T15 mean profile shows, if anything, more depleted 18O values compared to the T13** record (Fig. 6b).' Is this negative difference significant (see below d-excess)? If it is significant, does this mean that post-deposition, at this site, is characterized by a decrease of d18O values? What process could be responsible of this

decrease?

O\_\_\_\_ 'Specifically for South Pole conditions (annual-mean temperature $-50°$C, accumulation rate 84mmw.eq.yr$-1$, surface wind speed 5ms$-1$), the firn isotopic composition showed annual-mean enrichment by firn ventilation after several years of $\sim$3‰ (Town et al., 2008).' In as much as the first cycle of T15 reflects undisturbed isotopic cycle ( -44 to -52‰ and annual average of -48‰ ), the annual average value after post-deposition (between 70 and 150 cm depth) is indeed enriched (-45 to -46‰ ) by nearly 3 ‰

O\_\_\_\_ 'For the first overlapping annual cycle, T15 exhibits an average difference from T13** of $-1.6$‰ for the other annual cycles the averages are $-0.4$, $\pm0$ and $-0.1$‰.' There seems to be an increase in values with depth, with the difference between T15 and T13** getting closer to zero. Is that trend significant?

\_\_\_\_i\_\_\_\_ The T13 profile, and its derivatives (T13* and T13**) do not show this trend. If it is significant, it could mean that this trend is a result of a post-deposition process yet unknown, that could also be responsible for the overall depletion of T15 relative to T13 (or T13*, T13**). This process would be oriented, and would bring preferentially light isotopes to the top and/or heavy isotopes to the bottom.

\_\_\_\_i\_\_\_\_ Qualitatively, sublimation (Sokratov and Golubev, 2009) is unlikely to produce this result; it would instead bring enrichment in the top layers. Oriented diffusion is also unlikely, because when it is active in summer, vapor moves downward, and would bring light isotopes to the bottom.

\_\_\_\_i\_\_\_\_ The ventilation process as described by Town et al. (2008) could contribute to this trend: Town et al. (2008) show that the winters become more and more enriched after burial, at least until the influence of the wind becomes null (40 cm). Looking at Figure 6b, there seems to be indeed a trend toward higher winter values when depth increases (especially in the original 'first 40 cm' located between 60 and 100 cm depth).

_____i_____ Regarding the summers values, they are too low for the first two summers (T15 relative to T13*) and too high for the next (deeper) summers. This could be explained by ventilation too. The summers at shallow depth are first depleted because of condensation of 'winter' vapor during the winters. But later on, they can be enriched again by 'summer' vapor entering during subsequent summers. Since more vapour is available in summer, this influence would become preponderant when layers are buried more deeply. (In winter the atmospheric air would contain only little vapour that would condensate quickly/entirely in shallow layers and not reach these deeper layers).

_____i_____ Of course all of this is very theoretical as long as we ignore the vapor isotopic composition in the atmosphere, and the direction of air fluxes.

O______ 'We note that the RMSD corresponding to the first value is above our stated detection limit.' See above («1‰

O______ 'Furthermore, the difference curve (Fig. 6b) does not show any clear seasonal timing which might be expected for a systematic post-depositional modification.' This affirmation could be nuanced. The maximum deviations (from zero) generally occur in phase with the extremums. The only case where the maximum deviation is not in phase (in front of the T15 extremums) is when the two curves T13** and T15 are not in phase with each other (110-120 cm) probably due to linear compression.

«<Figure 6b: annotated»> See attached figure (Figure 2)

O______ 'We nevertheless note the possibility that post-depositional changes by wind-driven firn ventilation are present at Kohnen Station but that their effect is unexpectedly weak and thus masked by the stratigraphic noise level.' See above («1‰

O______ 'Finally, we note the small tendency towards negative values of the differences between the T15 and T13** mean profiles (Fig. 6), What do you mean by 'negative tendency'? Is it the increase with depth or just the average of the differences between T15 and T13**?

O_____ '. . .we cannot reject the null hypothesis that both spatial and residual temporal differences originate from the same distribution,' This sentence is unclear, could you be more explicit ?

O_____ On Figure 7b, the 'spatial' difference and 'temporal' difference seem to have the same mean value (which seems negative). Did you made a test to evaluate if the average value is statistically different from zero, for the two variables? The fact that they 'originate from the same distribution' does not really prove that the average value is null for both, just that their averages are not statistically different from each other.

O_____ Is the negative difference between T15 and T13** significant? (See above). If it is the case, then there is a contradiction between the two tests. If not, this negative difference cannot be used as an argument to select processes.

O_____ On Figure 7b, the 'spatial' difference appears to have wider distribution than the 'temporal' difference. Does your statistical test include the width of the distribution?

O_____ 'the histogram of the temporal differences is even more symmetric than for 18O.' This clearly supports the absence of new deposition processes. Is there a trend with depth for the d-excess values?

O_____ '(1) Seasonal variation and intermittency of precipitation cause the discrepancy between isotope and local temperature data (Sime et al., 2009, 2011; Persson et al., 2011; Laepple et al., 2011).' This hypothesis could have come earlier (in the introduction or when the discrepancy was described).

O_____ 'At Kohnen Station, a large part of the annual accumulation is assumed to occur in winter since little or no precipitation is observed in the summer field seasons. However, the exact seasonal and inter-annual variation of accumulation is still unclear due to the lack of sufficiently precise, year-round observations (Helsen et al., 2005).' Idem

/

[Figure]

/

/

CONCLUSIONS

O______ 'The trench records show a pure downward-advection of the isotope signal within the open-porous firn, further influenced only by firn diffusion and densification, with no evidence for substantial additional post-depositional modification.' This conclusion is largely supported by the data, and the statistics. Quantitatively, the remaining difference can be accounted for by spatial noise, and thus there is no proof of another process active (and no need for it). Qualitatively, ventilation may still be happening.

O______ 'Year-long isotope studies (e.g. in seasonal intervals) focusing on the near-surface would help to constrain isotope modifications at the interface of surface snow and atmosphere.' Yes, more field campaigns, especially at this interface are acutely needed to understand what is happening.

/

/

/

/

/

/

/

_________________Technical comments__________________________

p8 line 7: 'T15-2 profile'.

p10, l16: 'deviations especially remain' remove 'especially'

p10, l30: "variability" miss an 'a'

p12, l28: 'occured during the 2 years' misses a 'r'.

p12, l3: 'modified T13' which one? Is it T13* like in the previous sentence or T13** like on the figure 6b? If it is T13**, could you also check the previous sentence, and give RMSD for T13** (for consistency)?

P16, l2: verify 'focussing'

p 16, l8: 'averaging' needs a second 'a'

Figure 2. The labelling is too small for longitude, latitude, and for the core and trench names. Is it possible to add the general wind direction?

Figure 5.

O_____ 'For each parameter set of compression and diffusion, we record the minimum root-mean square deviation of the profiles (contour lines) for the optimal downward-advection value (colour scale).' From this legend it seems that only the (diffusion; compression) couples were tested (while in the main text it seems that all the parameters are varied independently). Could you clarify this point? Is the downward advection the parameter with the less impact on RMSD? This is suggested by not treating the parameters equally in this figure.

––––––––––––––––––––––––––––––––

[Figure]

**Fig. 1.**

[Figure]

**Fig. 2.**

---

## Referee Comment (RC2) · Anonymous Referee #2 · 8 May 2017

The manuscript "Constraints on post-depositional isotope modifications in East Antarctic firn from analysing temporal changes of isotope profiles" by Thomas Munch and co-authors is devoted to the study of post-depositional changes of snow isotope composition in central Antarctica using the huge dataset of recently obtained data. The authors clearly demonstrated, using robust statistical methods, that the observed evolution of the vertical profiles of snow isotopic composition can be explained without significant influence of the post-depositional processes. In general, I enjoyed reading the manuscript and suggest that it may be published as it is, or with minor corrections.

I think the authors could slightly modify the main idea of the conclusion of the manuscript. In the current version they state "no evidence for substantial additional

post-depositional modification", meaning that they do not expect post-depositional modifications stronger than 1 per mil for oxygen 18. Indeed, 1 per mil is a very small value comparing to the spatial variability due to the stratigraphic noise. But on other hand, if considering the post-depositional modifications of the whole annual snow layer, 1 per mil is rather big value – it's an equivalent of about 1.25 *C of air temperature change! Thus, the obtained results still give some room for the post-depositional modifications of the snow isotopic composition, although they are less than 3 per mil as expected from the modeling (Page 14). Other comments or corrections: Figure 1 would be more informative if you add a wind rose, or just an arrow showing the prevailing wind direction. Page 14, line 11, "Sublimation led in lab studies…" – the sentence looks somewhat awkward, please consider revision. Page 16, line 8: averaging Page 16, line 10: did you want to say that the spatial separation should be well above the spatial decorrelation length?

---

## Author Comment (AC1) · 24 May 2017

Reply to the Review Comments of Anonymous Referee #1

on the manuscript

TC-2017-35: Constraints on post-depositional isotope modifications in East Antarctic firn from analysing temporal changes of isotope profiles

by Thomas Münch et al.

*We are very grateful for the enormously careful and thorough review of our manuscript and for the many detailed and constructive comments that will help to improve the work. Below there is a point-by point response to both the general and all specific comments raised by the referee. The original referee comments are set in normal font, our answers (author comment, AC) are typeset in blue.*

_______________________General comments_____________________
This article presents new measures of isotopic compositions (d18O, d-excess) in the first 2 meters of snow at Kohnen (Antarctica). These measurements are used to evaluate how the isotopic signal is modified with time (over a two-year interval), after deposition, at this site. The authors also present a simple model including 3 post-deposition processes, and use it to simulate the evolution of d18O values for the same period of time. The model and data results are coherent with each other. The authors conclude that no other processes (besides these three) are necessary to account for d18O evolution in the snow layers. Besides this study of post-deposition, the authors compare the spatially averaged d18O profile in the snow to measured temperature evolution (AWS) and note a strong discrepancy. Since post-deposition processes do not explain this discrepancy, they propose that processes before or during deposition have to be investigated.

I recommend that this paper be accepted with moderate revisions.

1) The data presented here are crucially needed at the moment. They not only represent a huge amount of field work and analysis, but also respect a carefully designed set-up to ensure the quality of the signal retrieved by minimization of horizontal noise. Such high-quality data are exactly what is required to evaluate quantitatively the impact of post-deposition processes.

2) The quantitative evaluation of the three processes studied through minimization of RMSD is clear, and the magnitudes obtained are coherent with independent estimates.

3) However, the articulation between the strategy of the field experiment and the broader issue of the discrepancy between interannual temperatures and interannual d18O could be more detailed in Introduction.

4) The authors could nuance their conclusion that post-deposition processes are unable to produce the interannual variability of d18O observed. Only three processes have been evaluated quantitatively, the others are rejected based only on qualitative observations (and are still subject of research).

AC:
We are happy that the referee considers our data and the work to be significant and important. Nevertheless we also acknowledge the mentioned generally weaker points of our manuscript. We will revise the work to improve the introduction (improving the elaboration of strategy of field work vs. broader issue of discrepant inter-annual temperature and d18O variations + including the discussion of precipitation intermittency (see specific answers below)). Further, we will tone down our conclusions to account for the stated detection limit of additional post-depositional changes (see also our reply to Anonymous Referee #2) and for the qualitative nature of some part of our results.

_______________________________Specific comments____________________

ABSTRACT

O_____ 'Here we reject the hypothesis of post-depositional change within the open-porous firn beyond diffusion and densification.' This sentence is unclear. Is it possible to use affirmative form?
AC:
We apologize that the sentence was not sufficiently precise. We will rephrase it to: "Here, we investigate the importance of post-depositional processes within the open-porous firn and find that further modifications besides those arising from diffusion and densification are unlikely."

O_____ 'These results show that the discrepancy between local temperatures and isotopes most likely originates from spatially coherent processes prior to or during deposition, such as precipitation intermittency or systematic isotope modifications acting on drifting or loose surface snow.' Why did you choose to evaluate post-deposition processes and not precipitation intermittency in this study? The latter is a strong candidate for the observed discrepancy. Is it due to a lack a measurements?
AC:
Yes, it is indeed the lack of measurements that prevents a quantitative evaluation of the precipitation intermittency. Over the year, measurements of accumulation are only available from the automatic weather station, the data however are strongly influenced by noise due to dune movements and snow drift. Snow stake measurements are only obtained in summer (thus, only record annual mean accumulation rates or at most the summer snowfall over the relatively short periods of the field seasons) and in addition are not available over the complete period of the trench records. Finally, the reliability of reanalysis precipitation amounts is unclear. We think that all these information are too detailed for the abstract. However, we will add a summary to the introduction in order to motivate why we focus our study on post-deposition.

INTRODUCTION

O_____ When you say that diffusion and condensation 'only smooth and compress the original signal', you should precise that you are talking about vapor diffusion against isotopic gradients.
AC:
It is indeed a good point to precise to which diffusion process we refer here. However, to our knowledge the term "against isotopic gradients" is not common in the literature. Diffusion rather acts "down" the (concentration) gradients. We will change the sentence to "The isotope ratios of buried snow are affected by firn densification (…) and by diffusion of interstitial water vapour driven by gradients in the isotopic composition (…)".

O_____ 'In contrast, the low local annual accumulation rates and potential seasonal intermittency of precipitation increase the time the surface is exposed to the atmosphere (Town et al., 2008; Hoshina et al., 2014) and therefore to processes that might alter the snow's original isotopic composition.' The intermittency of precipitation does not only favor post-deposition processes through exposition to the atmosphere; it can also shape the d18O signal because of irregular accumulation.
AC:
This is of course correct. We will add the discussion of precipitation intermittency to this paragraph (p.2, ll. 10-27) of the introduction.

O_____ 'These processes can act either on loose snow in the post-condensational phase (falling or drifting snow), . . .' Could you precise which processes are active then? It is not wind redistribution, since these processes have to be spatially coherent.
AC:
Falling or drifting snow at the surface might be already influenced by fractionation due to sublimation and condensation processes, similar to deposited surface snow as mentioned later in this paragraph of the introduction. We agree with the reviewer's apparent impression that the logical order of the paragraph is not optimal and we will revise the entire paragraph to improve this.

O_____ 'This discrepancy stresses the importance of contributions other than regional temperature alone to the formation of the isotope signal. /// In this study, we investigate whether post-depositional isotope modifications in the open-porous firn contribute to the observed discrepancy between isotopes and local temperature at Kohnen Station.' This transition is very short. Could you indicate briefly what are the other contributions and why this study is dedicated to post-deposition?
AC:
We will provide a link here to the processes discussed in the first part of the introduction (see comments above) and then explain why we now focus on post-deposition (basically since we lack precise measurements to evaluate precipitation intermittency, see also our comments above).

Figure 1.

O_____ Do you have information on precipitation amounts over this period? Or on summer d18O in the snowfall? Does the summer d18O in the snowfall follow the evolution of summer temperatures? If precipitation amounts are unknown, please state it here, not later in the Discussion. . . It will be easier to understand why you focus on post-deposition processes.
AC:
No, unfortunately we do not have information on precise precipitation amounts over the period of the trench data as explained in our reply to your second comment on the abstract. We also have no precise information on seasonal timing of precipitation, only the qualitative observation from the field seasons that there is little accumulation in the sommer months. For this reason, we also lack systematic measurements of summer snowfall which could be compared to temperature observations. We will add a summary of these information to the introduction in order to motivate why we focus the manuscript on post-deposition (see above).

O_____ '. . .we have designed our study such that it allows for the first time to quantitatively follow the isotopic changes and thus to test for post-depositional effects over a time span of 2 years.' What do you expect for the evolution of the variability over 2 years? An attenuation or an amplification? If you expect only an attenuation, then post-deposition is obviously not responsible for the discrepancy between temperature and d18O interannual variations (attenuating a flat profile will not lead to increased variability). If you expect amplification, then why do you simulate only 'attenuating' post-deposition processes?
AC:
This is a very good point. However, we do not expect only attenuating effects. Post-depositional processes depend on other climatic features than temperature alone, such as wind speed, time the surface layer is exposed to the influence of the atmosphere, radiation, humidity, surface topography, etc., and could imprint these features to the isotopic signal in the firn. For inter-annual variations of these climatic features one would then also expect post-depositionally driven inter-annual variability of the isotopes. We will add these ideas to the manuscript at the end of the 2nd paragraph of the introduction.
In addition, we simulate the known influences of downward-advection, diffusion and densification – processes which are certainly at play and of which only diffusion is attenuating – not because we only expected attenuating effects but to disentangle the effects of these three processes from any further post-depositional effects.

RESULTS

Table 2.

O_____ 'The higher variances in vertical direction of the T15 records are partly expected for autocorrelated data in combination with a larger record length,' It seems also stronger for the horizontal variability. Do you have an explanation for that? There is also a strong increase of the signal-to-noise ratio. Does it mean that the mean profile in 2013 is less well known?
AC:
We will add the confidence intervals of the variance estimates (using the effective degrees of freedom to account for the autocorrelation of the data). This shows that the variance estimates are not

significantly different from each other. The uncertainty of the signal-to-noise ratio estimates is given by one standard error. The different trench valus are hence also likely not significantly different. We will add the relevant information to the table caption.

Figure 3.

O_____Considering only the part of the profiles that is complete, there seems to be an increase of d18O with depth. The shallowest winter (24 cm) has a very low value compared to the deepest winter (153 cm). There is a similar trend for summers (-37‰ for the summer at 173 cm and -44‰ for the summer at 33 cm). Is it possible to test this trend with a linear regression? Do you have information on the continuation of this trend at greater depths? If this trend is verified, what process could be responsible of such an increase?

AC:

Yes, testing on trends is of course possible. We tested the average T15 trench summer maxima and winter minima for a linear trend. The seeming increase in isotopic winter minima is not strongly significant against depth nor time ($p = 0.1$). In contrast, the trend in isotopic summer maxima is significant both against depth and time ($p < 0.01$). This trend cannot be explained by summer temperatures (see Figure 1), but it could be caused by changes in other climatic parameters such as the amount of summer snowfall. For greater depths, we only have preliminary data from our trench campaign which however do not show any continuation of the summer trend.

Figure 4.

O_____It is really difficult to compare quantitatively the two curves on this figure, because they are not superposed. Could you put them on the same d18O scale, and shift the 2013 curve 'optimally'?

«<Figure 4: superposed»> see attached figure (Figure 1)

AC:

We do not agree with the reviewer on this point. It is certainly correct that the visual comparison of the profiles would be improved by superposing the plots. However, putting the plots on top of each other using only one y axis could visually imply that the profiles originate from the same expedition which is not the case. For that reason we decided to offset the plots vertically with respect to each other and use separate y axes (we noticed however that both axes do not have the exact same scale; we will change this to facilitate the comparison). Putting the plots on the same y axis and in addition using the optimal shift, as suggested by the reviewer, would preempt part of our results at this point of the manuscript. By contrast, our aim here is to show both mean profiles on their original depth scale, and from this point on discuss the different processes (downward-advection, diffusion, densification) that finally lead to Figure 6b. In summary, we suggest to leave this part as it is but leave it to the editor to decide on this issue.

O_____ 'In the 2 years, the T13 isotope profiles are advected downwards, compressed by densification and smoothed by firn diffusion.' Attenuation is not very clear here. There is attenuation between 75 and 120 cm depth (blue zone). However, between 60 and 75 cm depth and also between 125 and 150 cm depth the profile after two years (2015) has larger amplitude (red zones). Adding attenuation to the initial d18O profile from T13 would increase the agreement in the blue zone, but decrease the agreement in the red zones.

AC:

We admit that this part was ambiguous and thank the reviewer for pointing towards that. The cited sentence was not intended to express a result but the expectation that these three processes are at play and must be quantitatively investigated first before one can assess the significance of further post-depositional changes. We will rephrase the sentence to make this clear: "Within the 2 years, we expect that the T13 profiles are advected downwards, compressed by densification and smoothed by firn diffusion."

Figure 5

O_____ 'For the downward-advection, we apply vertical shifts between $\Delta = 40$ and 60 cm,' This range is too large to stay within the bounds of the first winter minimum

(47-53 cm would be enough) and too small to permit the shifting of the curve by one cycle (shift of 25-75 cm required). How is it possible that 60 cm become an optimum (it should lead to anti-correlation)?

AC:
The referee's estimates are totally valid and correct but are based on the trench data. However, to find the optimal parameter set of advection, diffusion and densification, we want to be as independent of the trench data as possible and therefore choose the values of vertical shifts accordingly. We will add this argument to the manuscript. A spaciously choice for the possible downward shifts are those values that cover the estimated range of annual accumulation rates observed in the wider vicinity of Kohnen Station (20-30 cm), as given in section 2.4. We see no motivation for further narrowing or enlarging the range of tested downward-advection values.

A vertical shift of the T13 profile by 60 cm can in fact be locally an 'optimal value' but only in combination with strong diffusion and densification. Shifting the profile by 60 cm alone indeed leads to anti-correlation and a high deviation from T15 (rmsd > 3 ‰), see attached Figure A1. However, for the combination of a large diffusion length (8 cm) and a strong compression (10 cm), this shift leads to the smallest possible deviation from T15 (rmsd ~1.3 ‰, upper right corner of Figure 5) since the strong diffusion essentially flattens the profile and the strong compression counteracts most of the anti-correlation that results from the vertical shift alone (Figure A1).

[Figure]

**Figure A1: The T13 mean profile shifted alone by 60 cm (red), and shifted by 60 cm, diffused by a differential diffusion length of 8 cm and compressed by 10 cm (blue), as compared to the T15 mean profile (black).**

O_____ Compression higher than 6 cm or diffusion length higher 4 cm leads to RMSD higher than 'doing nothing' (1.05 at the point of origin). This is interesting as it gives an upper bound for the impact of these processes. It also confirms the estimates from independent datasets.

AC:
Yes, this is absolutely correct. It is also mentioned in the text that the optimal parameter values found from varying the parameters across their ranges are close to the ones that we obtained fully independently from the trench data (p. 10 ll. 10-14).

Figure 6

O_____ 'We obtain the best agreement (RMSD = 0.92 ‰, Fig. 5; r = 0.93) between the T15 and the modified T13 mean profile (= T13*) for the optimal parameters $\Delta_{opt}$ = 50.5 cm, $\sigma_{opt}$ = 2.3 cm and $\gamma_{opt}$ = 3.5 cm (Fig. 6).' Even if adding attenuation generally increases agreement with 2015, is it really the best scenario to apply here (considering red zones)? If the diffusion length was computed only on the zone where attenuation is evident (between 75 and 120 cm) would it have the same value?

AC:
This is a very good point. Of course it is possible that the best-possible fit we obtained does not represent the correct physical processes, thus is right for the wrong reasons. However, the agreement of

the parameter values between our best-fit and the independent estimates argues against this possibility. To compute the differential diffusion length σ only for the "blue zones" certainly yields a different result as the one found in the manuscript but this value would (1) be more uncertain since fewer data are used for the estimate, and (2) it would represent a subjective choice based on knowledge of the trench data. We consider it important to reach our statistical conclusions with the least possible amount of data-based presuppositions.

O_____ Did you try to move the profile of T13 vertically (more or less enriched in heavy isotopes) to get a better fit? Of course the processes tested here would not lead to a change in the mean value, but it could give information on other processes (maybe for discussion).

AC:
We repeated the analysis for Figure 5 looping over different isotopic mean shifts of the T13 mean profile (from -1 to +1 ‰ in steps of 0.1 ‰). Indeed, shifting the mean value of the T13 mean profile results in a further reduced RMSD with the T15 mean profile (Figure A2). We find a new minimum RMSD for a shift in mean value of -0.4 ‰. The associated optimal parameter values of downward-advection, diffusion and densification are with $\Delta_{opt}$ = 50.5 cm, $\sigma_{opt}$ = 2.4 cm and $\gamma_{opt}$ = 3.4 cm equal or similar to the ones obtained without shifting the mean. However, we have no possible explanation for such a change in mean value, but still we think that this finding is worth adding to the discussion and thank the referee for raising the issue.

[Figure]

**Figure A2: The RMSD between the T13\* and the T15 mean profile as a function of the shift in the mean of the T13 profile.**

O_____ Could you give us an estimation of the attenuation due to diffusion? It could be useful for future comparisons (to other data or models). Roughly from the graph (T13\*), the half-attenuation seems to be of ~0.6 ‰ and the initial half-amplitude of about 2.2 ‰ which would correspond to a quite strong attenuation, of the order of 25 % over two years. What would be the attenuation in the 'blue zone': 75-120 cm depth?

AC:
Many thanks for these estimates. We have attached the Figure A3 showing the typical exponential decline of the seasonal amplitude with depth according to the local depth-dependent diffusion length (Kohnen Station parameters) and a seasonal cycle with wavelength of 25 cm (range of 20—30 cm). At 1m depth, the seasonal amplitude has been reduced to ~ 75—85 % of its initial value at the surface, at 1.5m depth the reduction is ~ 60—80 %. These results are well captured by your rough estimate from the data.

[Figure]

**Figure A3: The relative decline with depth in seasonal amplitude due to diffusion for Kohnen Station parameters for the upper 5 m of firn. The black line shows the amplitude reduction for a seasonal cycle with a wavelength of 25 cm, the shading gives the reduction for wavelengths from 20 to 30 cm (lower to upper bound).**

O_____ p10 l20: 'can be seen' and l21: 'clearly': It would be easier to see the improvement if there were somewhere a figure showing T13 (unmodified) and T15 superposed. Without this figure, the term 'seen' should be avoided/replaced.
AC:
We agree that in this part it might be a bit hard for the reader to follow our statements. However, all the necessary information is contained in Figure 5 – just shifting T13 optimally is represented by the lower left corner of the figure, adding diffusion and densification improves the RMSD towards the black dot. We will add these information to the manuscript to guide the reader more carefully. In addition, we will rephrase the sentences to avoid the terms 'can be seen' and 'clearly'.

O_____ 'Nevertheless, both processes play a significant role in explaining part of the temporal changes. This can be seen if we only shift the T13 mean profile vertically to find the maximum correlation with T15. . .' Is the RMSD of 'only compression' different from the one of T13*? How much improvement is obtained by adding the diffusion to the 'compression only' experiment?
AC:
Yes, according to Figure 5, the minimum RMSD of 'only compression' is 0.98 ‰ (Figure 5, along the vertical axis the minimum RMSD is found for a compression value of 3 cm). Thus, adding diffusion improves the match slightly by 0.06 ‰. This is a small value but nevertheless we think that diffusion explains at least some part of the temporal differences between T13 and T15, especially when taking into account that the diffusion model near the surface could very likely be further improved by accounting for different diffusion lengths for summer and winter layers (see discussion). However, this approach is beyond the scope of our study. In any case, we will weaken the statement at this point of the manuscript to reflect that the gain in RMSD by adding diffusion is small (which is also the case for adding compression alone).

O_____ 'deviations especially remain around the isotopic extreme values, in particular for the first overlapping cycle and the depths around 100 and 125–140 cm.' As expected, the deviation after post-deposition is high mostly in the red zones (first cycle, 125-140 cm), where the amplitude in 2015 is larger than the amplitude in 2013. For these zones adding diffusion leads to higher deviations than doing nothing (and the term 'remaining difference' is maybe not the best adapted).
AC:
Yes, it is correct that our model increases the deviations in some parts of the profile overlap, however, in total it minimizes the root mean square deviation and thus is overall still the best-case scenario. We will improve the discussion of our results at this point to take that into account, especially we will emphasise that the partially increased profile deviations are expected since some T13 amplitudes were already initially smaller than for T15, and thus diffusion cannot lead to an improved profile match here.

We will thus replace the inappropriate phrase 'remaining differences'.

O_____ What do you call 'extreme values'? All the extremums? Or only the summer at
175 cm and the winter at 70 cm? If you are talking about the extremums, then there is
a contradiction with p. 14: 'Furthermore, the difference curve (Fig. 6b) does not show
any clear seasonal timing. . .'
AC:
This is a good observation, thank you for pointing towards that. Indeed, our statement here was too
generalized and thus in contradiction to the statement on p. 14. We will weaken the statement and
rewrite the sentence to replace the term 'especially remain at'.

O_____ 'This gives a best shift of 48.5 cm, but clearly the agreement is less pronounced
(RMSD = 1.1‰ r = 0.88) compared to. . .' On the Figure 5, at the point of
origin (no diffusion, no densification), the RMSD is 1.05. It reaches 0.92 ‰ for optimal
compression and diffusion. Thus these two processes are a step in the right direction,
but finally do not improve the RMSD very much.
AC:
We agree that the gain in RMSD is not particularly high when adding diffusion and densification (see
alos comments above). As a consequence, we will weaken our statement here by stating 'This gives a
best shift of 48.5 cm, but the agreement is slightly less pronounced (…)'.

O_____ p. 10, l25: 'Taking these processes into account leads to a good match of the
trench mean profiles (Fig. 6b). However, deviations on the order of 0.9–1‰ remain.'
What were the deviations before taking them into account?
AC:
We agree that it is a good idea to guide the reader more carefully here. We will change the sentence in
order to reflect that just accounting for downward-advection already yields a good match of the
profiles, which is further improved slightly by adding diffusion and densification (with a gain in
RMSD by ~0.15 ‰).

O_____ 'These can have two causes: firstly, additional temporal changes driven by unaccounted
post-depositional processes;' Could you precise what other processes you
are thinking about? Or maybe just make a note toward the section where these unaccounted
processes are discussed? Listing possible processes could help to research
specific features expected in the remaining variability.
AC:
We refer here to the processes discussed in the introduction (post-deposition such as sublimation and
ventilation) and will add a link to emphasise that.

O_____ 'secondly, remaining spatial variability since we average a large but finite num-
ber of records which do not originate from the exact same position.' It seems coherent
to evaluate the remaining variability as spatial noise, if this variability is random. How-
ever, it may not be the case here (slight trend toward higher values with depth, see
below).
AC:
This is a very good point. However, we see no indication for *not* assuming that the spatial variability is
random, thus white noise. Firstly, the slight increase of the difference curve towards higher values with
depth (Figure 6b) is not significant (see below). Secondly, diffusion smoothes the noise and thus leads
to autocorrelation, meaning that even white noise shows autocorrelation after diffusion. In other words,
the autocorrelation of the difference curve hence does not invalidate the white noise assumption.
In addition, we use the statistical noise model as a first test to assess whether the differences between
T13*/T13** and T15 can be explained by spatial variability. This test indeed assumes white noise
between the trenches. However, as a second test we use the formal statistical KS test which yields the
same result as the first test but makes no assumption about the coherence of the noise between the
trenches.

O_____ 'The agreement of both estimates indicates that the remaining profile differences
between the modified T13 mean profile and T15 (Fig. 6b) can be entirely explained
by spatial variability through stratigraphic noise. We note however that the
squared RMSD lies at the upper end' If there is still a doubt in your mind after the

mathematical demonstration, why do you use the term 'entirely' in the first sentence?
This term also seems in contradiction with the end of the paragraph. To facilitate reading,
you could add a layer of uncertainty such as: 'At first order, the agreement of both
estimates indicates'
AC:
Thank you for spotting this inconsistency in our language. We will remove the word 'entirely' and add
a layer of uncertainty to our statement, as suggested.

Figure 7

O_____ 'We find that the distributions of the spatial differences between the mean
profiles of each trench campaign (T13–1 vs. T13–2 and T15–1 vs. T15–2, Fig. 7a) are
statistically indistinguishable' Could you explicit the results of these tests with simple
words? What is the more general conclusion of this first test? That the sampling
strategy has no influence on the results? That the uncertainty is the same for T13 and
T15?
AC:
We note that the cited phrase 'distributions (…) are statistically indistinguishable' is the formal and
correct statement for the obtained result. In more simple words it means that there is no significant
difference in spatial variability between each trench pair, thus between the two seasons. Alternatively,
one can state that the spatial variability estimates from the trench pairs belong to the same underlying
distribution (regarding mean and variance / location and width). We will add a more thorough
explanation to the manusript to facilitate the interpretation of the result.

O_____ 'More importantly, the combined distribution of spatial variability is also indistinguishable
from the distribution of the temporal differences between the T15 and the
modified T13 mean profile' Does this test evaluate if the difference between T13** and
T15 is more than just the difference between T13** and T15 that comes from having a
different location?
AC:
This test evaluates if the (temporal) differences between T13** and T15 belong to the same underlying
distribution as the (combined) spatial differences. Thus, the null hypothesis of the test is that the
differences between T13** and T15 just arise from the fact that the trenches have a different location,
thus that the temporal differences can be explained by spatial variability alone. We find that we cannot
reject this null hypothesis. We will rephrase this part of the manuscript in order to facilitate the
interpration of our results (in line with our last comment).

O_____ How do you 'combine' spatial differences between trenches? The distances
considered are not exactly the same (~350 m between T13-1 and T13-2; ~500 meters
between T15-1 and T15-2; ~200 m between the mean T13 position and the mean T15
position). Do you apply a weighting by distance?
AC:
No, we do not apply a weighting by distance. This is motivated by the fact that the change in spatial
(horizontal) correlation of the stratigraphic noise is large in the first metres, but only small or even zero
for larger distances (> ~10 m) (Münch et al., 2016). Thus, it does not matter if the trenches are
separated by 300, 400 or 500 m. This is in fact also underpinned by the first KS test: we find that the
spatial variability between the trench pairs belong to the same underlying distribution (see above), thus
do not depend on the distance between the trenches. We will add this information to the manuscript at
this point.

DISCUSSION

________Densification, diffusion and stratigraphic noise________

O_____ 'We found a strong resemblance...' This 'strong resemblance' is largely
brought by moving downward the profile (advection). The impact of compression and
diffusion, even if it is significant, is still very small.
AC:
Yes, this is correct. We will take this into account by emphasising that the major portion of the
agreement achieved by our model is a result of the downward-advection.

O\_\_\_\_\_ 'our assumption of a linear profile compression with depth is certainly a rough approximation given the actually observed seasonal firn density variation (Laepple et al., 2016).' In what direction would that process intervene? Preferential compression of summers or winters?
AC:
This is a good question and worth mentioning in the manuscript. In Figure 6 in Laepple et al. (2016), there seems to be stronger densification (change in density) of summer compared to winter layers. However, the short data do not allow to estimate if this is a robust feature. In general, the seasonality of densification in Antarctic firn is very unclear (Laepple et al. 2016 and references therein).

O\_\_\_\_\_ 'In detail, the diffusion correction improves the match of the trench mean profiles in the medium depth range but also results in higher deviations of the profile minima at the top and bottom part of the overlap (Fig. 6).' This observation is much welcome but should have come earlier in the manuscript, when the deviations are first described.
AC:
Yes, absolutely. We wil change the manuscript accordingly as already outlined in our above answers to the comments relating to the same issue.

O\_\_\_\_\_ 'Part of this mismatch might be reduced by accounting for the seasonally varying firn temperature resulting in stronger (weaker) diffusion for summer (winter) seasons (Simonsen et al., 2011).' How exactly? Does this mean that summers would be more attenuated than winters (due to stronger attenuation when they are still at the surface)? What about temperature gradients? They might not only favor attenuation, but also redistribute heavy and light isotopes vertically.
AC:
Yes, indeed summers would be more attenuated than winters due to the higher surface temperatures, especially close to the surface where the difference in diffusion length between summer and winter is largest (see Figure 1c in Simonsen at al., 2011). We will enhance the discussion regarding this point. The effect of temperature gradients is subject of open research.

\_\_\_\_\_\_\_\_\_\_\_\_\_\_Additional post-depositional modifications\_\_\_\_\_\_\_\_

O\_\_\_\_\_ '. . .any additional post-depositional changes of the isotopic composition of the firn, below 10 cm, must be on average clearly below the residual stratigraphic noise level, thus « 1‰'' Thus the change can be of more than 1 ‰ as long as it goes on opposite directions at top and at bottom (the average being zero)?
AC:
No, this is not correct. The limit of 1 ‰ stated here refers to 1 ‰ RMSD. The RMSD is independent of the sign of the actual differences, thus opposite changes at top and bottom with the average being zero would still result in a non-zero RMSD (e.g., changes of +1 ‰ at the top and -1 ‰ at the bottom would result in a RMSD of sqrt(2) ~ 1.4 ‰, larger than the limit set by the stratigraphic noise level.) We will add the therm 'RMSD' to the cited statement in order to clarify this ("thus « 1‰ RMSD").

O\_\_\_\_\_ 'This conclusion is also supported by comparing the qualitative nature of the differences between the mean profiles (Fig. 6b)' Regarding this difference (violet curve): is it possible to add the zero line, to discriminate between positive and negative differences? Is it possible to add the difference T15-T13 (with optimum downward advection), to see where the post-deposition has been most effective?
AC:
This is a good idea. We will add the zero line and the second difference curve for the T15-T13 differences accounting only for the optimal downward-advection.

O\_\_\_\_\_ 'the T15 mean profile shows, if anything, more depleted 18O values compared to the T13** record (Fig. 6b).' Is this negative difference significant (see below d-excess)? If it is significant, does this mean that post-deposition, at this site, is characterized by a decrease of d18O values? What process could be responsible of this decrease?
AC:

A t-test accounting for the autocorrelation of the data shows that the overall mean difference (~ -0.45 ‰) is not significantly different from zero (p = 0.4). We will add this information to the manuscript. In consequence, we refrain here from discussing potential mechanisms that might be responsible for an overall negative difference since the differences are more likely just a random expression of the diffused stratigraphic noise.

O_____ 'Specifically for South Pole conditions (annual-mean temperature −50°C, accumulation rate 84mmw.eq.yr−1, surface wind speed 5ms−1), the firn isotopic composition
showed annual-mean enrichment by firn ventilation after several years of ~3‰
(Town et al., 2008).' In as much as the first cycle of T15 reflects undisturbed isotopic
cycle ( -44 to -52‰ and annual average of -48‰ ), the annual average value after postdeposition
(between 70 and 150 cm depth) is indeed enriched (-45 to -46‰ ) by nearly
3 ‰
AC:
Please see our author comment below (marked by an asterisk *).

O_____ 'For the first overlapping annual cycle, T15 exhibits an average difference from
T13** of −1.6‰ for the other annual cycles the averages are −0.4, ±0 and −0.1‰·'
There seems to be an increase in values with depth, with the difference between T15
and T13** getting closer to zero. Is that trend significant?
AC:
The trend is not strongly significant (p = 0.12) when accounting for the autocorrelation of the data. We will add this information to the manuscript.

____i____ The T13 profile, and its derivatives (T13* and T13**) do not show this trend.
If it is significant, it could mean that this trend is a result of a post-deposition process
yet unknown, that could also be responsible for the overall depletion of T15 relative to
T13 (or T13*, T13**). This process would be oriented, and would bring preferentially
light isotopes to the top and/or heavy isotopes to the bottom.
AC:
Thank you for these considerations. Although the trend is not significant (see above), it is indeed an interesting observation which calls for further studies.

____i____ Qualitatively, sublimation (Sokratov and Golubev, 2009) is unlikely to produce
this result; it would instead bring enrichment in the top layers. Oriented diffusion
is also unlikely, because when it is active in summer, vapor moves downward, and
would bring light isotopes to the bottom.
AC:
Thanks once more for these thoughts. We will consider to add these discussion points to the manuscript, depending on how well they can be incorporated into the present discussion.

____i____ The ventilation process as described by Town et al. (2008) could contribute
to this trend: Town et al. (2008) show that the winters become more and more enriched
after burial, at least until the influence of the wind becomes null (40 cm). Looking at
Figure 6b, there seems to be indeed a trend toward higher winter values when depth
increases (especially in the original 'first 40 cm' located between 60 and 100 cm depth).

____i____ Regarding the summers values, they are too low for the first two summers
(T15 relative to T13*) and too high for the next (deeper) summers. This could be
explained by ventilation too. The summers at shallow depth are first depleted because
of condensation of 'winter' vapor during the winters. But later on, they can be enriched
again by 'summer' vapor entering during subsequent summers. Since more vapour is
available in summer, this influence would become preponderant when layers are buried
more deeply. (In winter the atmospheric air would contain only little vapour that would
condensate quickly/entirely in shallow layers and not reach these deeper layers).

AC(*):
Combined response to the last two comments as well as the one not answered above (*):

Thank you very much for these detailed considerations. You are partially right but mix two different

observations. Indeed there seems to be a trend in T15 winter layers towards less negative values over depth. You are also correct in saying that the Town et al. ventilation study also shows enrichment of winter layers over time. However, the conclusion that the observed T15 winter trend over depth can thus be explained by post-deposition following Town et al. must assume that initially all winter layers looked like the first layer of T15 (or at least had similar initial minimum values). We have no means to validate this assumption. All we can do is directly compare the winter layers of T13* to T15 which are direct counterparts of the same season. If we only look at the first winter layer which was closest to the surface at the time of excavation of T13, thus presumably being under strongest influence of the atmosphere, we see a rather strong depletion of the layer after 2 years (the layer in T15 is more negative than the layer of T13* by about 2 ‰). This is just the opposite of what is suggested by Town et al.! Also the other winter layers either show no change (2nd and 3rd layer) or also more depleted values after 2 years.

Regarding the summers: Your observation is right that here indeed the change of the first three summer layers appears consistent with the Town et al. results. However, looking at annual mean differences, again we find no support for the possibility of ventilation (as stated in the manuscript, the first annual cycle is overall more depleted after 2 years by 1.6 ‰, not enriched as suggested by ventilation, since the winter layers don't show the "expected" strong enrichment overcompensating the summer depletion). In addition, we see no motivation for the hypothesis suggesting that ventilation at Kohnen Station should be only active in summer.

In summary, we see no clear support for post-depositional changes from firn ventilation that would be in line with the Town et al. study. However, of course we cannot directly reject the hypothesis since the effect might still be there but just too small to be detected by our analysis. Therefore, we will elaborate the discussion of the difference curve in more detail, as outlined here, and also attenuate our conlusions.

_____i_____ Of course all of this is very theoretical as long as we ignore the vapor isotopic composition in the atmosphere, and the direction of air fluxes.
AC:
Yes, indeed. That is why we also suggest in our final conclusions that future studies should combine measurements and analyses similar to ours with measurements of the atmospheric vapour isotope composition.

O_____ 'We note that the RMSD corresponding to the first value is above our stated detection limit.' See above («1‰
AC:
We are sorry but we do not fully understand what you mean with your comment.

O_____ 'Furthermore, the difference curve (Fig. 6b) does not show any clear seasonal timing which might be expected for a systematic post-depositional modification.' This affirmation could be nuanced. The maximum deviations (from zero) generally occur in phase with the extremums. The only case where the maximum deviation is not in phase (in front of the T15 extremums) is when the two curves T13** and T15 are not in phase with each other (110-120 cm) probably due to linear compression.
«<Figure 6b: annotated»> See attached figure (Figure 2)
AC:
We do not fully agree with these statements. This is almost true if one considers only the extreme values in general. If one looks in more detail (see attached figure A4, which is Figure 6b from the manuscript with the zero line added) only 3 out of 5 winter minima in T15/T13[*/**] isotope values coincide with a minimum in the difference curve, and only 2-3 out of 5 summer maxima coincide with a maximum in the difference curve. The remaining extremes in both cases either coincide with the opposite extreme in the difference curve or with a difference of around 0 ‰. For us it is thus reasonable to conclude that the 'difference curve does not show any clear seasonal timing'. We rather think that the difference curve appears to exhibit some kind of seasonality due to the smoothing effect of diffusion. However, the shortness of the data does not allow a formal test of this hypothesis. We will modify the discussion at this point of the manuscript to elaborate on our reasoning.

[Figure]

**Figure A4: Same as Figure 6b from the manuscript but with zero line added on the difference curve (T15-T13\*\*).**

O\_\_\_\_\_ 'We nevertheless note the possibility that post-depositional changes by wind-driven firn ventilation are present at Kohnen Station but that their effect is unexpectedly weak and thus masked by the stratigraphic noise level.' See above («1‰
AC:
We are sorry but we do not fully understand what you mean with your comment.

O\_\_\_\_\_ 'Finally, we note the small tendency towards negative values of the differences between the T15 and T13\*\* mean profiles (Fig. 6), What do you mean by 'negative tendency'? Is it the increase with depth or just the average of the differences between T15 and T13\*\*?
AC:
We meant the seeming (thus insignificant, see above) increase of the differences with depth between T15 and T13\*\*. We will rephrase the text to clarify this point.

O\_\_\_\_\_ '. . .we cannot reject the null hypothesis that both spatial and residual temporal differences originate from the same distribution,' This sentence is unclear, could you be more explicit ?
AC:
We apologize that this sentence is unclear. However, it is the correct expression of the result. Nevertheless we will rephrase/amend the respective passage to clarify the statement. Basically it means that all residual temporal differences are very likely attributable to spatial variability alone, and hence no further post-depositional processes need to be invoked to explain the mean and amplitude of the differences.

O\_\_\_\_\_ On Figure 7b, the 'spatial' difference and 'temporal' difference seem to have the same mean value (which seems negative). Did you made a test to evaluate if the average value is statistically different from zero, for the two variables? The fact that they 'originate from the same distribution' does not really prove that the average value is null for both, just that their averages are not statistically different from each other.
No, we haven't included this information in the manuscript. However, applying the t-test, taking into account the autocorrelation of the data, we find that the average differences are statistically not different from zero (p = 0.4). We will add this information to the manuscript.

O\_\_\_\_\_ Is the negative difference between T15 and T13\*\* significant? (See above). If it is the case, then there is a contradiction between the two tests. If not, this negative difference cannot be used as an argument to select processes.
AC:
As given in our above answer, the average difference between T15 and T13\*\* is not significant and there is thus no contradiction between the two tests.

O_____ On Figure 7b, the 'spatial' difference appears to have wider distribution than the 'temporal' difference. Does your statistical test include the width of the distribution?
AC:
Yes, the KS test is sensitive to differences in both location (mean) and width (variance) of the empirical distribution functions of both samples. We will add this information to the Methods section 2.5.

O_____ 'the histogram of the temporal differences is even more symmetric than for 18O.' This clearly supports the absence of new deposition processes. Is there a trend with depth for the d-excess values?
AC:
No, the difference curve for the d-excess values (T15 – T13**) does not show a significant trend with depth (p = 0.4, accounting for autocorrelation). The average difference is, as for d18O, also not significantly different from zero (p = 0.9, accounting for autocorrelation).

O_____ '(1) Seasonal variation and intermittency of precipitation cause the discrepancy between isotope and local temperature data (Sime et al., 2009, 2011; Persson et al., 2011; Laepple et al., 2011).' This hypothesis could have come earlier (in the introduction or when the discrepancy was described).
AC:
As we stated in our replies to the referee comments on the introduction, we will discuss the effect of precipitation intermittency on the isotope signal already in the introduction.

O_____ 'At Kohnen Station, a large part of the annual accumulation is assumed to occur in winter since little or no precipitation is observed in the summer field seasons. However, the exact seasonal and inter-annual variation of accumulation is still unclear due to the lack of sufficiently precise, year-round observations (Helsen et al., 2005).'
Idem
AC:
As described above, also the fact of our imprecise knowledge on seasonal and inter-annual variations of precipitation and accumulation will be included already in the introduction.

CONCLUSIONS

O_____ 'The trench records show a pure downward-advection of the isotope signal within the open-porous firn, further influenced only by firn diffusion and densification, with no evidence for substantial additional post-depositional modification.' This conclusion is largely supported by the data, and the statistics. Quantitatively, the remaining difference can be accounted for by spatial noise, and thus there is no proof of another process active (and no need for it). Qualitatively, ventilation may still be happening.
AC:
We agree with the referee that our results still leave room for post-depositional changes – either with very small magnitudes so that their effect is "masked" by stratigraphic noise in our analysis, or occurring directly at the surface where we do not have sufficient data to assess this possibility. In consequence, we will weaken our statement here.

O_____ 'Year-long isotope studies (e.g. in seasonal intervals) focusing on the near-surface would help to constrain isotope modifications at the interface of surface snow and atmosphere.' Yes, more field campaigns, especially at this interface are acutely needed to understand what is happening.
AC:
We fully agree with you.

_________________Technical comments_____________________

p8 line 7: 'T15-2 profile'.
AC:
We are not sure what you mean here. Reading the entire paragraph, we noticed that "T15-x profile" or "T15-x mean profile" has not been used in a consistent fashion. We will replace all occurrences of "T15-x profile" by "T15-x mean profile" throughout the text.

p10, l16: 'deviations especially remain' remove 'especially'
AC:
We will remove the word 'especially'.

p10, l30: "variability" miss an 'a'
AC:
The typo will be corrected.

p12, l28: 'occured during the 2 years' misses a 'r'.
AC:
The typo will be corrected.

p12, l3: 'modified T13' which one? Is it T13* like in the previous sentence or T13** like
on the figure 6b? If it is T13**, could you also check the previous sentence, and give
RMSD for T13** (for consistency)?
AC:
 '(…) modified T13 mean profile'  here indeed referred to T13*. However, for consistency we will also
mention the squared RMSD value for T13** and will rephrase the sentence as follows:
"For comparison, the square of the RMSD between the T13* (T13**)  and T15 mean profile is 0.85
(0.88) (‰)$^2$. The agreement of the estimates indicates that (…)"

P16, l2: verify 'focussing'
AC:
We will change this part to "Year long istotope studies (…) *with a focus on* (…)"

p 16, l8: 'averaging' needs a second 'a'
AC:
The typo will be corrected.

Figure 2. The labelling is too small for longitude, latitude, and for the core and trench
names. Is it possible to add the general wind direction?
AC:
We will increase the label fonts and add an arrow indicating the mean wind direction (57°).

Figure 5.

O_____ 'For each parameter set of compression and diffusion, we record the minimum
root-mean square deviation of the profiles (contour lines) for the optimal downward-advection
value (colour scale).' From this legend it seems that only the (diffusion;
compression) couples were tested (while in the main text it seems that all the parameters
are varied independently). Could you clarify this point? Is the downward advection
the parameter with the less impact on RMSD? This is suggested by not treating the
parameters equally in this figure.
AC:
We apologize for the fact that the figure caption has been misleading. In fact, we independently varied
all three parameters (downward-advection, diffusion, compression) according to the given ranges, as
stated in the text. However, in this figure we show the results projected onto the optimal advection
values, thus for each pair of diffusion length and compression value we only show the minimum
RMSD from varying across the range of advection values. This choice was necessary in order to be
able to display the three-dimensional parameter space in only two dimensions. In fact, the downward-
advection has the largest influence on the RMSD. We will improve the caption description in order to
clarify these points.

References:

[1]     Laepple, T., Hörhold, M., Münch, T., Freitag, J., Wegner, A., and Kipfstuhl, S.: Layering of
        surface snow and firn at Kohnen Station, Antarctica: Noise or seasonal signal?, J. Geophys.
        Res. Earth Surf., 121, 1849–1860, doi:10.1002/2016JF003919, 2016.
[2]     Münch, T., Kipfstuhl, S., Freitag, J., Meyer, H., and Laepple, T.: Regional climate signal vs.

local noise: a two-dimensional view of water isotopes in Antarctic firn at Kohnen Station, Dronning Maud Land, Clim. Past, 12, 1565–1581, doi:10.5194/cp-12-1565-2016, 2016.

[3]     Simonsen, S. B., Johnsen, S. J., Popp, T. J., Vinther, B. M., Gkinis, V., and Steen-Larsen, H. C.: Past surface temperatures at the NorthGRIP drill site from the difference in firn diffusion of water isotopes, Clim. Past, 7, 1327–1335, doi:10.5194/cp-7-1327-2011, 2011.

---

## Author Comment (AC2) · 24 May 2017

Reply to the Review Comments of Anonymous Referee #2

on the manuscript

TC-2017-35: Constraints on post-depositional isotope modifications in East Antarctic firn from analysing temporal changes of isotope profiles

by Thomas Münch et al.

*We thank the referee for carefully reviewing our manuscript and for the constructive comments that will help to improve it. Below there is a point-by point response to both the general and the specific comments raised by the referee. The original referee comments are set in normal font, our answers (author comment, AC) are typeset in blue.*

The manuscript "Constraints on post-depositional isotope modifications in East Antarctic firn from analysing temporal changes of isotope profiles" by Thomas Munch and co-authors is devoted to the study of post-depositional changes of snow isotope composition in central Antarctica using the huge dataset of recently obtained data. The authors clearly demonstrated, using robust statistical methods, that the observed evolution of the vertical profiles of snow isotopic composition can be explained without significant influence of the post-depositional processes. In general, I enjoyed reading the manuscript and suggest that it may be published as it is, or with minor corrections.

I think the authors could slightly modify the main idea of the conclusion of the manuscript. In the current version they state "no evidence for substantial additional post-depositional modification", meaning that they do not expect post-depositional modifications stronger than 1 per mil for oxygen 18. Indeed, 1 per mil is a very small value comparing to the spatial variability due to the stratigraphic noise. But on other hand, if considering the post-depositional modifications of the whole annual snow layer, 1 per mil is rather big value – it's an equivalent of about 1.25 *C of air temperature change! Thus, the obtained results still give some room for the post-depositional modifications of the snow isotopic composition, although they are less than 3 per mil as expected from the modeling (Page 14).

AC:

We totally agree with the reviewer that our results make post-depositional changes in addition to diffusion and densification unlikely at our study stite, but still leave room for such effects of the order of <1 ‰, and also very close to the surface where our data are insufficient. But we have to bear in mind that this limit of 1 ‰ refers to the root mean square deviation of the T15 and T13 profiles (after accounting for diffusion and densification) calculated over the entire record's overlap, thus on the seasonal scale. If we consider annual means, this value should be much smaller. However, still we will tone down our conclusions by stating that additional post-depositional changes appear unlikely, but that we can only constrain this to changes down to the order of less than ~1 ‰ RMSD on a seasonal basis.

Other comments or corrections:

Figure 1 would be more informative if you add a wind rose, or just an arrow showing the prevailing wind direction.

AC:

(We assume that Referee #2 refers to Figure 2 here). We will add a wind rose and an arrow indicating the prevailing wind direction (57°) to this plot.

Page 14, line 11, "Sublimation led in lab studies. . ." – the sentence looks somewhat awkward, please consider revision.

AC:

We apologize for the fact that this sentence was poorly formulated. We will rephrase it as follows: "In lab studies it was shown that sublimation leads to isotopic enrichment (Sokratov and Golubev, 2009); the modelling of post-depositional modification as a result of wind-driven firn ventilation by Town et al. (2008) yielded overall annual-mean enrichment from the enrichment of isotopic winter layers."

Page 16, line 8: averaging
AC:
This typo will be corrected.

Page 16, line 10: did you want to say that the spatial separation should be well above the spatial decorrelation length?
AC:
No, indeed we mean well *below* the decorrelation length of the stratigraphic noise. If you compare two isotope profiles that are spaced above the decorrelation length, the contribution of stratigrahic noise to the overall variability of the profiles will be different between the profiles since the noise is spatially no longer correlated. As a consequence, the resulting spatial variability between the profiles will likely mask any temporal changes you want to detect. By contrast, for a spacing below the decorrelation length, the noise contributions will show high similarity and therefore it will be easier to discriminate temporal and spatial variability. The downside of such an approach is that you have to make the second measurement as close as possible to the first one, making disturbances or contaminations of the second profile by the previous measurement(s) more likely. In the manuscript, for the sake of clarity, we will amend the cited sentence as follows: "Alternatively, single records can only be compared faithfully for temporal changes when their spatial separation is well below the spatial decorrelation length of the stratigraphic noise, minimising the amount of spatial variability between the records."

---

## Editor Comment (EC1) · J. Savarino (Editor) · 27 May 2017

O_____When you say that diffusion and condensation 'only smooth and compress the original signal', you should precise that you are talking about vapor diffusion against isotopic gradients. AC: It is indeed a good point to precise to which diffusion process we refer here. However, to our knowledge the term "against isotopic gradients" is not common in the literature. Diffusion rather acts "down" the (concentration) gradients. We will change the sentence to "The isotope ratios of buried snow are affected by firn densification (...) and by diffusion of interstitial water vapour driven by gradients in the isotopic composition (...)".

EC: Sorry but I don't understand the statement "by diffusion of interstitial water vapour

driven by gradients in the isotopic composition (...)". It appear to me that the sentence is claiming that isotopic gradient is a driving force of change. Vapour diffusion in snow is driven by T gradients not by isotopic composition. The isotopic gradient will drive the diffusion only if the system under consideration was isothermal, purely diffusional. In the present situation, the change of enthalpy induced by the T gradient is orders of magnitude greater than the change of enthalpy induced by the isotopic gradient. Isotopic composition change is thus a result, not a driving force. Please rephrase so that the reader is not confused by which process is responsible for the change in the isotopic composition.

EC: Regarding the shift of the curves in fig4. I will suggest to keep the original plot. Eventually, the curves can be x-axis shifted for taking into account the accumulation between the two samplings but no superposition is required in my view.

Please submit your revised MS for approval.

---

## Author Comment (AC3) · 31 May 2017

We thank the editor for his comments for which our author comments (AC) are given below.

O_____When you say that diffusion and condensation 'only smooth and compress the original signal', you should precise that you are talking about vapor diffusion against isotopic gradients. AC: It is indeed a good point to precise to which diffusion process we refer here. However, to our knowledge the term "against isotopic gradients" is not common in the literature. Diffusion rather acts "down" the (concentration) gradients. We will change the sentence to "The isotope ratios of buried snow are affected by firn densification (...) and by diffusion of interstitial water vapour driven by gradients in the

isotopic composition (...)".

EC: Sorry but I don't understand the statement "by diffusion of interstitial water vapour driven by gradients in the isotopic composition (...)". It appear to me that the sentence is claiming that isotopic gradient is a driving force of change. Vapour diffusion in snow is driven by T gradients not by isotopic composition. The isotopic gradient will drive the diffusion only if the system under consideration was isothermal, purely diffusional. In the present situation, the change of enthalpy induced by the T gradient is orders of magnitude greater than the change of enthalpy induced by the isotopic gradient. Isotopic composition change is thus a result, not a driving force. Please rephrase so that the reader is not confused by which process is responsible for the change in the isotopic composition.

AC:

We thank the editor for clarifying this issue. It is indeed true that only for isothermal firn diffusion is driven alone by the different isotopic composition of the layers. For non-constant temperatures, the main driver of diffusion are temperature gradients since the temperature directly affects the vapour concentration above the ice and thus also the concentration of heavy and light isotopologues in the vapour. In this case, the diffusion does not necessarily lead to a pure smoothing of the isotopic composition in the firn. This is probably also what referee #1 referred to in the original comment. We overlooked this fact in our answer since we approximate the effect of diffusion in our study assuming isothermal firn even at the trench depth scale. We will add a respective note to the revised manuscript in section 2.4 where the diffusion model is described in order to clarify this and the fact that pure smoothing only occurs for isothermal firn. Regarding the introduction, since in polythermal firn the effect of diffusion may not be a pure Gaussian-like smoothing and not only driven by different isotopic composition of the layers, we suggest to rephrase the cited sentence to a more general statement: "The isotopic composition of buried snow and firn is affected by diffusion of interstitial water vapour (...) and by densification (...); however, these processes do not lead to a

net change in the isotopic composition."

EC: Regarding the shift of the curves in fig4. I will suggest to keep the original plot. Eventually, the curves can be x-axis shifted for taking into account the accumulation between the two samplings but no superposition is required in my view.

AC:

Thank you for your comment on this. We will keep the original plot in the revised mansucript.